# Comparing Redox and Intracellular Signalling Responses to Cold Plasma in Wound Healing and Cancer

Adrian I. Abdo [1,2,*] and Zlatko Kopecki [3,*]

1    Richter Lab, Surgical Specialties, Adelaide Medical School, University of Adelaide, Adelaide, SA 5000, Australia
2    Department of Surgery, The Basil Hetzel Institute for Translational Health Research, The Queen Elizabeth Hospital, Woodville, SA 5011, Australia
3    Future Industries Institute, STEM Academic Unit, University of South Australia, Mawson Lakes, SA 5095, Australia
*    Correspondence: adrian.abdo@adelaide.edu.au (A.I.A.); zlatko.kopecki@unisa.edu.au (Z.K.)

**Abstract:** Cold plasma (CP) is an ionised gas containing excited molecules and ions, radicals, and free electrons, and which emits electric fields and UV radiation. CP is potently antimicrobial, and can be applied safely to biological tissue, birthing the field of plasma medicine. Reactive oxygen and nitrogen species (RONS) produced by CP affect biological processes directly or indirectly via the modification of cellular lipids, proteins, DNA, and intracellular signalling pathways. CP can be applied at lower levels for oxidative eustress to activate cell proliferation, motility, migration, and antioxidant production in normal cells, mainly potentiated by the unfolded protein response, the nuclear factor-erythroid factor 2-related factor 2 (Nrf2)-activated antioxidant response element, and the phosphoinositide 3-kinase/protein kinase B (PI3K/Akt) pathway, which also activates nuclear factor-kappa B (NFκB). At higher CP exposures, inactivation, apoptosis, and autophagy of malignant cells can occur via the degradation of the PI3K/Akt and mitogen-activated protein kinase (MAPK)-dependent and -independent activation of the master tumour suppressor p53, leading to caspase-mediated cell death. These opposing responses validate a hormesis approach to plasma medicine. Clinical applications of CP are becoming increasingly realised in wound healing, while clinical effectiveness in tumours is currently coming to light. This review will outline advances in plasma medicine and compare the main redox and intracellular signalling responses to CP in wound healing and cancer.

**Keywords:** cold plasma; plasma-activated water; redox signalling; cancer; wound healing; Nrf2; MAPK; PI3K/Akt; endoplasmic reticulum stress

## 1. Introduction

Plasma is an ionised gaseous mixture of excited ions, free radicals, and electrons that emits an electromagnetic field and ultraviolet light. Thermal plasma, existing in the universe as stars, fire, lightning, and auroras, uses energy in the form of heat to strip gases of electrons to create plasma in thermal equilibrium. In contrast, cold atmospheric-pressure plasma, referred to simply as "cold plasma" (CP) from here, is a partially ionised plasma, where intense electric fields (kHz–MHz range) are applied to low temperature gas (noble gas, air, $O_2$, and $N_2$, a mixture of either dry or humidified) to liberate electrons at high energy, and hence temperature ($10^4$–$10^5$ °C), leading to inelastic and elastic electron impact collisions that partially ionise the gas with a degree of ionization of 0.001% or lower [1,2]. The heavy gas particles are unaffected by the electric field and remain cool, while the much smaller and fewer highly kinetic and hot electrons that impact and ionise the gas do little to change the heavy gas particle temperature, meaning that the CP is in a non-equilibrium thermodynamic state and the overall resulting temperature can be below 40 °C [3]. As a result, CP is tissue-tolerable, usually up to several minutes of continuous application before

significant tissue heating occurs, and enables diverse medical applications of CP [3]. This also means that when the electric field (i.e., power) is disabled, the high-energy electrons are rapidly quenched by the gas and plasma ceases to form, making the use in medicine highly controllable. Therefore, CP has been tested in a variety of medical applications to birth the still evolving field of plasma medicine.

There is a vast diversity in CP reactor design. In plasma medicine, CP is most commonly produced using corona, microwave, gliding arc, dielectric barrier discharge (DBD), and piezoelectric direct discharge (PDD) CP. The DBD consists of two parallel electrodes, of which one electrode is of a dielectric material (glass, quartz, ceramics, rubber, plastic, enamel, and teflon) [4]. Biological tissue is also dielectric and can act as the second electrode in medical applications of DBD plasmas [5]. The corona discharge CP is a very weak discharge with a low electron density in the plasma plume that forms around the tip of one electrode (usually fitted as a wire or needle) [6,7]. On the other hand, the gliding arc plasma is a quasi-thermal plasma with a much higher electron and ion density at the smallest distance (1–5 mm) between two diverging electrodes [8–10]. The gliding arc plasma is then propelled by high gas flow or buoyancy, 'gliding' along the diverging electrodes, which increases the volume of the plasma, decreasing the discharge intensity to a lower temperature (generally < 80 °C) non-equilibrium state before a new gliding arc is discharged [8,9]. More recently, the PDD configuration has entered plasma medicine, where the discharge originates directly from a resonant piezoelectric transformer that can transform voltages greater than 1000-fold [11]. The main advantage of the PDD is that the compact solid state piezoelectric transformer requires little power, which significantly miniaturizes the CP device for more versatile use [11]. The designs of these CP devices are more comprehensively reviewed elsewhere [2,4,10,11]. For a point target application of CP within millimetres, plasma jets are produced using one of the aforementioned CP discharge types, which are then propelled into a narrow jet by a secondary non-ionised gas. For treatment of a larger area, DBD, corona, and gliding arc discharges can cover larger areas; these plasmas have chaotically shifting discharges that create "dead zones" causing inefficient plasma coverage. Meanwhile, multiple plasma jets can be constructed in an array for simultaneous treatment of a larger area [12,13]. This plasma jet array can have a relatively consistent and uniform plasma profile over the target area and can use jet–jet interactions to amplify and produce unique plasma plumes by adjusting the discharge gas flow rate [12–16]. Consequently, many findings from CP technology studies may be specific to particular CP designs. Therefore, it is important to note that some findings discussed in this review should not be generalised to all CP designs.

The medical use of CP is generally centred around the milieu of reactive oxygen and nitrogen species (RONS) that are formed [17]. For direct CP exposure, highly reactive and short-lived RONS, including hydroxyl ($^{\bullet}OH$), hydroperoxyl ($^{\bullet}OOH$), superoxide ($O_2^{\bullet-}$) and nitric oxide ($NO^{\bullet}$) radicals, oxygen (O) and nitrogen (N) atoms, singlet oxygen ($^1O_2$), hydrogen ($H^+$), hydroxyl ($OH^-$), and peroxynitrite ($ONOO^-$) ions are abundantly present [18–20]. These highly reactive species have a profound effect on cell membrane structure, porosity, hydrophobicity, and fluidity, mainly through lipid and protein oxidation, which compromises the function of the cell that also affect downstream intracellular signalling [21]. When applied for longer exposure times, the RONS produced can cause significant potent redox imbalances that hinder cell proliferation or lead to destruction of cells, which is beneficial to prevent tumour regrowth [22–24]. In contrast, lower CP exposure times can promote cell proliferation, increased motility/migration, activate inflammatory signalling pathways, and antioxidant production in healthy skin and immune cells, which is an essential response in wound healing and tissue regeneration [2,25]. The difference in the application of CP between wound healing and cancer killing revolves around 'hormesis' [21], where "low doses" of CP can be harmful when applied to malignant tissue, but can also promote positive wound healing effects in healthy tissue inducive to tissue regeneration. Here, it must also be recognised that "low" and "high" dose with respect to CP exposure in this review do not refer to a quantitative dose of CP in this context, but instead

refer the use of CP on hormesis principles. The first attempts at defining a standardised "plasma dosage" were performed by varying the CP treatment duration, voltages, flow rate, and feed gas composition, culminating in an expression for CP dosage as $D \sim QVt$ ($D$: entire plasma dose applied to the cells, $Q$: discharge gas flow rate, $V$: output voltage, and $t$: treatment time) [26]. Unfortunately, this definition of plasma dose is overly simplistic; it only accounts for a few linear factors, while several other factors are absent, including nonlinear factors that make plasma dosage determination complicated [27]. This led to the equivalent total oxidation potential (ETOP), which is based on the oxidation potential of the reactive species in the plasma discharge, UV, electric fields, and the interactions between these factors [28]. However, there are still various limitations to the ETOP, with refinement needed before there is a possibility of implementation and standardisation [2].

Plasma-activated liquids (PAL), most commonly plasma-activated water (PAW) and plasma-activated medium (PAM), are an indirect method of applying CP. PAL are produced either by direct discharge of CP into the liquid (e.g., water, media), or positioning CP discharges in the gas phase above the liquid to facilitate reactions at the gas–liquid interface [18,29]. While PAL is usually applied several minutes to up to several months after initial CP activation, the oxidative potential of CP is "stored" in the liquid for later use. Hence, several long-lived secondary RONS are present instead of highly reactive and short-lived RONS, including ozone ($O_3$), nitrate/nitric acid ($NO_3^-$/$HNO_3$), nitrite/nitrous acid ($NO_2^-$/$HNO_2^-$), and hydrogen peroxide ($H_2O_2$) [2,20,30]. Particularly $H_2O_2$, $NO_3^-$, and $NO_2^-$ will linearly accumulate in PAL as a function of CP treatment duration and energy density discharged per liquid volume and $NO_2^-$ and $H_2O_2$ decay while $NO_3^-$ continually forms post-plasma activation under acidic conditions [31,32]. These molecules also decrease the pH of the liquid, creating an antimicrobial environment capable of effectively killing bacteria [33–36] and eliciting various redox-dependent cellular signalling responses in eukaryotic and cancerous cells. Additionally, PAL have been shown to exhibit the desired biological effects when used after the short-lived species are degraded post-plasma activation, indicating that PAL could be storable for future use when sealed and stored at appropriate temperature [18,37–39]. Consequently, PAL has the advantages of being potentially storable, transportable, with no risk of heating tissue, and is more feasible for use in busy clinics.

Although plasma medicine was conceived at the end of the 20th century, a detailed analysis of mammalian cell signalling pathways responsive to plasma medicine has only recently been elucidated. This review aims to outline the current clinical use of plasma medicine and the cell signalling pathways responsible for eliciting cell responses to CP therapy in both activating cells, as seen in wound healing, and destruction of cancerous cells from in vitro, ex vivo, and in vivo studies. Although the antimicrobial effect of CP is well established and is an important part of wound care, this activity is independent of mammalian cell responses and is not discussed here, but has been covered by several other fantastic reviews [40–42]. Finally, gaps in future research and the limitations of plasma medicine are discussed.

## 2. Plasma Medicine in the Clinic

Plasma medicine, both CP and PAL, have diverse applications in various biomedical fields, including broad-spectrum antimicrobial therapy, anticancer therapy, acute and chronic wounds (e.g., diabetic foot ulcers, haemostasis) therapy, and dentistry [2,18,30,40,41,43–52]. Several studies have examined the use of CP for disinfection of burn injuries [53], as well as for CP activation of hydrogels to enable the delivery of antimicrobial drugs or CP-generated RONS [54,55]. Currently, there are a few Conformité Européenneor (CE; European Conformity)-certified class IIa/b CP medical devices, including the kINPen® MED plasma jet (Neoplas Med, Greifswald, Germany), SteriPlas® microwave plasma torch (Adtec, Twickenham, UK), PlasmaDerm® DBD device (CINOGY, Duderstadt, Germany), Plasma Care® (Terraplasma Medical, Garching, Germany), and CPT®cube/CPT®patch DBD device (ColdPlasmaTech, Greifswald, Germany). To date, a number of clinical case

studies have demonstrated benefits of CP therapy together with standard of care for difficult-to-treat wounds, including flap donor sites with exposed tendons [56] and complex wounds following cranio-maxillo-facial surgery [57]. These studies highlighted CP as the reliable, minimally invasive, easy-to-apply chairside treatment option resulting in effective healing without adverse events, which may be particularly useful for patients with impaired healing conditions due to the presence of systemic or local risk factors [56,57]. However, further studies are required to validate these observations in randomised controlled trials (RCTs) with larger patient populations and in comparison to control groups. Currently, clinical plasma medicine is constrained to wound healing, but clinical studies on cancer are rapidly emerging. Three hallmarks of clinical CP application to wounds include (1) antimicrobial efficacy, (2) modulation of redox signalling with subsequent stimulation of cell proliferation and migration properties via the release of growth factors, and (3) increased tissue oxygenation and micro-circulation, which is important during healing and angiogenesis. The two most studied areas of CP medical applications include wound healing and cancer; however, the absence of standardized guidelines across different CP treatments impedes the current reproducibility and effective assessment of CP treatment efficacy.

### 2.1. Wound Healing

In clinical wound management, plasma medicine and CP devices have been approved for therapeutic use because CP simultaneously has broad-spectrum antimicrobial activity, promotes wound healing, and regulates tissue inflammation [40,44,58]. This could potentially aid in treating chronic non-healing wounds, including diabetic foot ulcers, arterial and venous leg ulcers, and pressure injuries. These patients have an impaired wound healing response and immunodeficiency, resulting in high susceptibility to the development of chronic and recurrent bacterial and fungal clinical infections [59,60], which significantly increases the risk of amputation [61,62]. In Australia alone, over 420,000 people are affected by a chronic wound at any time, with health-care related costs exceeding AUD 3.5 billion, approximately 2% of national health care expenditure [63]. There is a desperate clinical need for novel therapeutic options, possibly using CP as adjuvant therapy to current standards in wound management. To address this, RCTs have shown CP in combination with best-practice care led to better outcomes.

In contrast to the above-mentioned case studies on complex wounds, the latest meta-analysis of CP therapy showed no significant improvement in wound healing (five studies, 148 participants) nor reduction in infections (four studies, 91 participants) in chronic wounds. However, exposure to CP also resulted in no adverse effects, suggesting relative safety in current protocols [64]. Since then, recent RCT has shown the efficacy of CP therapy in improving wound healing. Four RCTs, (i) 37 non-diabetic patients with non-healing wounds, (ii) 78 patients with mixed wound aetiology, and (iii–iv) 44 and 45 diabetic patients with chronic wounds, showed that wound closure was significantly accelerated, with improvements to wound-specific patient-reported metrics of quality of life and pain [47,48,65,66]. Another RCT in 44 diabetic patients revealed a greater reduction in systemic inflammatory cytokines and chemokines, including interleukin-1 (IL1), -8 (IL8), interferon-gamma (IFNγ), and tumour necrosis factor-alpha (TNFα) in the CP therapy group [49]. Most recently, preliminary results from the multicentre RCT Plasma On Chronic Wounds for Epidermal Regeneration (POWER) study on wound healing of uninfected, lower leg wounds in 47 patients showed that CP treatment resulted in ≥60% and ≥90% wound closure in 28% and 16% of participants, respectively, while standard wound therapy alone led to ≥40% wound reduction in 18% of subjects [67]. No adverse effects were observed, and antibiotic use was lower in the CP treatment group. The study is ongoing and expected to be completed by the end of 2024 (German Clinical Trials Register: DRKS00019943). Clinical studies on CP technology for wound healing are summarised in Table 1.

Another important aspect of the effects of CP on wound healing is the pro-angiogenic signalling of the wound bed to revascularize newly formed granulation tissue. To date, studies have shown that local vascular responses to CP include the promotion of platelet activation, aggregation, and fibrin polymerisation to accelerate blood coagulation and aid healing [68]. Short (5 min) CP exposure on intact skin and wounds has been demonstrated to increase tissue oxygenation and flow through localised promotion of endogenously produced NO (NO from CP cannot permeate the epidermis), a potent paracrine vasodilator and inflammatory regulator, to improve vascular circulation for over 1 h after CP therapy is stopped [69–72]. Later, (i) one study of 20 patients with diverse vascular morbidities, (ii) two studies with 30 combined healthy volunteers, and (iii) 20 male patients with diabetes-related chronic leg ulcers found a significant increase in cutaneous capillary circulation and subsequent blood and tissue $O_2$ saturation after treatment with the PlasmaDerm DBD CP device [73–75]. Although not confirmed in clinical trials, CP was suggested to result in phosphorylation (activation) of endothelial NO synthase (eNOS) and increased NO production. In murine wounds, this was shown to subsequently result in increased levels of vascular endothelial growth factor (VEGF) release and platelet-derived growth factor receptor-beta (PDGFRβ) expression to promote angiogenesis and tumour growth factor-beta 1 (TGFβ1) paracrine stimulation of collagen I deposition and wound re-epithelialization [76,77]. This supports the hypothesis that the paracrine-stimulating effect of CP therapy on subcutaneous vascular and epidermal tissues promotes wound healing and tissue regeneration. Future RCTs are imperative to optimise CP treatment duration and efficacy in developing personalised therapies. Understanding the exact mechanism underlying the effect of CP treatment on wound healing and investigating the combined impact of CP and the traditional standard of care is fundamental for enhancing and refining the efficacy of CP in wound management [78]. The potentially high cost of CP devices and treatments may limit accessibility for some patients in developing countries, while longitudinal studies should help assess the long-term effects and sustainability of CP treatment as adjuvant therapy in wound management.

**Table 1.** Phase I and II prospective clinical trials of CP technology for the promotion of wound healing.

| CP Device | Ref. | Study Design | Population and Treatment Groups | Primary Results |
|---|---|---|---|---|
| SteriPlas® plasma torch | [65] | RPCT ($n = 37$) | Chronic non-healing wounds. SWC with 2 min CP 1×/week (group 1, $n = 14$), 3×/week (group 2, $n = 13$), or placebo (1×/week, $n = 10$). | Wound area significantly reduced >60% with 1×/week treatment ($p = 0.005$). No additional benefit with 3×/week. |
| kINPen® Med plasma jet | [66] | MC, RCT, NIT ($n = 78$) | Chronic non-healing wounds. Equal groups CP treatment or SWC. CP treatment (30 s/cm² wound) 3× 1st week, 2× 2nd week, then 1×/week for 4 weeks. | Relative wound area significantly lower in CP-treated group by 5th visit. Lower infection frequency and significantly faster time to infection healing. |
| | [47] | SB, RPCT ($n = 45$) | DFU. Wound care with CP ($n = 33$) or placebo ($n = 32$) applied for 30 s/cm² wound once daily for 5 days, followed by 3 treatments every 2nd day. | CP treatment led to 26% greater wound closure in CP compared to placebo ($p = 0.03$). NSD in bacterial load. |

**Table 1.** *Cont.*

| CP Device | Ref. | Study Design | Population and Treatment Groups | Primary Results |
|---|---|---|---|---|
| Bioplasma Jet | [79] | RCT ($n = 42$) | Patients with pressure ulcers. Wound care with 1 min/cm$^2$ CP ($n = 23$) or without CP ($n = 19$). CP applied once weekly for 8 weeks. | Bacterial load reduced after 1 week of CP treatment. The wound size, wound base and exudate were significantly improved after two CP treatments. |
| PlasmaDerm® VU-2010 DBD | [80] | Two-armed, open, RCT ($n = 14$) | Chronic venous leg ulcers. CP ($n = 7$) or SWC ($n = 7$). CP (45 s/cm$^2$) ulcer, 3×/week for 8 weeks + 4-week follow-up | NSD in ulcer lesion size between groups. Significant increase in bacteria-free region ($p = 0.0313$). |
| Housemade He plasma jet | [48] | DB, RCT ($n = 44$) | DFU. Standard care with ($n = 22$) or without ($n = 22$) CP therapy. CP (5 min) 3×/week for 3 weeks. | CP accelerated wound closure after 3 weeks compared to control (*cf.* 61% vs. 21%, respectively, $p < 0.05$). Bacterial load significantly reduced by CP, but recolonised before the next CP treatment. |
| CPT®cube & CPT®patch | [67] | * MC, RCT ($n = 47$) | Uninfected chronic leg wounds. CP treatment (2 min) 3×/week for 4 weeks ($n = 25$) compared to SWC ($n = 22$). 3- and 6-month follow-up. | CP treatment resulted in ≥90% wound closure in 16% of participants; SWC alone only lead to ≥40% wound reduction in 18% of subjects. |
| PlasmaDerm® FLEX9060 DBD | [71] | C, cohort ($n = 20$) | Healthy volunteers with single 90 s application of CP to determine microvascular effects. | Oxygen saturation and perfusion significantly elevated in microcirculation up to 8 and 11 min after CP therapy, respectively. |
| | [73] | self-C, cohort ($n = 10$) | Healthy volunteers with single 4.5 min CP application to determine microvascular effects. | Oxygen saturation and perfusion significantly elevated after CP exposure over 60 min follow-up. |
| | [74] | C, cohort ($n = 20$) | Patients with chronic, infection-free leg wounds with single 1.5 min CP application to determine microvascular effects. | CP therapy significantly increased deep capillary perfusion after 11 min to end of 30 min follow-up. Superficial perfusion significantly elevated 1–6 min follow-up. |
| | [75] | C, cohort ($n = 20$) | Otherwise-healthy participants requiring skin grafting. CP treatment (90 s) at the skin donor site. | Deep capillary perfusion significantly increased entire 30 min follow-up to CP. Tissue oxygen significantly increased 1–5 min after CP therapy. |

Abbreviations: C: controlled, CP: cold plasma, SB: single-blinded, DB: double-blinded, RCT: randomised controlled trial, RPCT: randomised placebo-controlled trial, MC: multicentre, NIT: non-inferiority trial, DFU: diabetic foot ulcers, SWC: standard wound care, NSD: no significant difference. * Study ongoing.

## 2.2. Cancer

CP cancer therapy, or "plasma oncology", has mostly gone through limited phase I clinical trials, with some phase II trials still underway (Table 2). Studies to date have demonstrated a potential role of CP in cancer treatment, including cancer remission, with a focus on the effects of CP on RONS cancer cell impact, myeloid cells, immunogenic cancer cell death, and tumour response to altered tumour microenvironment [81]. A first-in-human trial in six patients with infected ulcerous squamous cell carcinoma (SCC), who received

thrice-weekly applications of CP therapy every other week, showed large patient-specific palliation against microbial infection [81]. More recently, 20 patients with stage IV solid tumours underwent CP therapy at the surgical margins following tumour resection to eliminate tumour regrowth [82]. No adverse effects or tumour recurrence were observed in patients with microscopic positive margins up to 32 months follow-up [82]. Although a primary culture of 9/10 patient tumour cells showed significantly reduced viability due to CP treatment in vitro, the apoptosis and oxidative stress pathways responded differently between patients [82], likely due to a variety of cancer types between patients. If the utility of CP is cancer type-dependent, more animal and clinical trials involving different malignant cancer types and tumour microenvironments are required to elucidate the real potential of CP in plasma oncology.

**Table 2.** Phase I and II prospective clinical trials of CP technology against cancers and tumours.

| CP Device | Ref. | Study Design | Population and Treatment Groups | Primary Results |
|---|---|---|---|---|
| kINPen® MED plasma jet | [81] | CS (*n* = 6) | Patients with oropharyngeal cancer received 1 min CP exposure 3× within 1 week. | Significantly improved palliation/QOL with reduced infection load. Too few participants to conclude anti-tumour effects. |
| | [83] | NCC, SB (*n* = 5) | Subjects previously exposed to 10–30 s CP in wounds. Followed up for 5 years to monitor tumour formation. | No suspicious or malignant lesions found. No pathological signs of tissue modification detected. |
| VIO3/APC3 electrosurgical plasma device | [84] | C, SA (*n* = 20) | Patients with cervical Intraepithelial Neoplasia (CIN) treated with CP 30 s/cm$^2$ with up to 24-week follow-up. | 19/20 subjects exhibited normal cervical histopathology. |
| | [85] | C, SA (*n* = 63) | | CP treatment group showed approximately double histological and cytological CIN remission rate, but NSD to control. |
| Canady Helios® plasma jet | [82] | MC, C (*n* = 20) | Patients with stage IV or recurrent solid tumour of mixed origins, with 21- to 32-month follow-up. CP treatment regime not disclosed. | No adverse effects or tumour recurrence. CP shows evidence of therapeutic effect as adjuvant to tumour resection. Evidence of pro-apoptotic protein expression induced by CP in cancer tissue biopsies. |

Abbreviations: C: controlled, CP: cold plasma, CS: case series, SB: single-blinded, NCC: nested case–control, SA: single-armed, MC: multi-centre, NSD: no significant difference, QOL: quality of life.

Recently, a phase II clinical trial in 63 patients with mild–moderate cervical intraepithelial neoplasia (CIN) tested a single 30 s CP application for prospective cervical cancer prevention compared to the control group of 287 participants that account for spontaneous remission [85]. A significantly higher remission rate was observed in CP-treated patients after both 3- and 6 months follow-up compared to controls, with fairly tolerable pain and discomfort [85]. RONS-mediated suppression of Akt (protein kinase B) phosphorylation (activation) and heat shock protein 27 (HSP27) with increased p53 phosphorylation/p53 binding protein expression promoting caspase 3/7-driven apoptosis has been indicated as a potential mechanism in a number of in vitro studies [84,86]. Unfortunately, the device used can rapidly (<5 s) heat tissue to damaging and painful temperatures (~80 °C) if not applied with constant uniform motion [84,86], unlike the other CE-certified CP devices, and hence may not technically be considered a CP device. Additionally, clinical characteristics and histological analysis between groups were significantly different due to the non-randomised trial design, making scientific conclusions highly tenuous. CP has also been used to treat late-stage head and neck SCC tumours, which experienced up to 80% reduction in tumour

surface, but unfortunately, in these clinical case studies, tumour growth relapsed [81]. On reflection by the investigators, this indicated that CP treatment alone may not be sufficient for management of advanced head and neck tumours, but a good adjunct therapy that has the potential to improve patients' quality of life (decreased infection load, enhanced social interaction) [87].

Several factors impede clinical trials of plasma oncology. First, plasma medicine has been limited to tumour cancers (currently excluding lymphomas and blood cancers), which are heterogeneous in classification, subtyping, and metastases [88,89]. Although CP has shown efficacy in osteosarcoma in vitro and in vivo, direct CP treatment is extremely invasive, and the use of less-invasive injectable PAL requires further research before clinical translation [90]. Fortuitously, CP treatment of blood from leukaemia patients has shown evidence that cancerous cells can be killed without adversely affecting haematological profiles [91]. This demonstrates that a hormesis approach between killing malignant cells and preserving healthy blood cells and (blood, not physical) plasma components may be viable, enabling the first steps towards clinical trials in non-tumour cancers. Secondly, it is unethical to trial CP without radiotherapy or chemotherapy, which can be patient-specific, and makes assessment of CP effect after accounting for interactions with other therapy extremely difficult. Third, some cancer patients may become desperate and join trials for the chance to be included in the treatment arm [92], introducing significant selection bias that may mask the true effectiveness of CP. This is not just theoretical, and has even been reported during a phase II trial in CIN patients who reported the "urgent desire to have children, where great fear and/or psychological stress" compelled them to join the study [84]. Similarly, studies might also intentionally forgo randomisation to allow for "maximum patient autonomy and voluntariness", which strongly influences the clinical characteristics between groups [85]. While these factors are justified on humane grounds, RCT design for plasma medicine in cancer is made considerably more difficult.

Given the relative infancy of plasma medicine and the advanced age of participants who usually qualify for recruitment in these studies, little is known about the long-term consequences of plasma medicine. One study with a five-year follow-up of five participants from a previous RCT found no long-term complications to overall health [83]. Unfortunately, although encouraging, this post hoc selection of participants for follow-up and a small sample size are extremely prone to systematic selection bias, which prohibits inference. Hence, prospective follow-up over longer time periods is required to confirm the long-term treatment effects.

## 3. Cold Plasma on Redox-Mediated Modification of Cell Components

Mammalian cells are enveloped in a fluid phospholipid bilayer with opposite-facing hydrophilic headgroups that contain cholesterol and proteins underneath the glycocalyx to form the cell membrane. The cell membrane allows nutrients, waste, and signalling molecules to be transported in and out of the cell. This transport is passive for hydrophobic and low-charge-density molecules, or active by protein channels or endo/exocytosis. Phospholipids like phosphatidylcholine, phosphatidylethanolamine, phosphatidylserine, and sphingomyelin bind to long-chain fatty acids containing double-bonds that are prone to oxidation [93]. Several studies have used phospholipid liposomes as models of mammalian cell membranes to show that short-lived RONS from CP, particularly O, $O_3$, $^{\bullet}OH$, $^1O_2{}^-$, $O_2{}^{\bullet-}$, and $NO^{\bullet}$ radicals, and $ONOO^-$ are strong oxidisers of fatty acid chains [94] that increase membrane porosity, consequently reducing membrane stability and increasing permeability [95–99], which can exacerbate further RONS entry directly into the cell. Furthermore, CP-derived RONS can form secondary lipid hydroperoxyl radicals (R-OO$^{\bullet}$) that can propagate damage to other cell compartments [94], leading to further damage after CP exposure has ceased [99]. Commonly, lipid peroxides, e.g., malondialdehyde and 4-hydroxy-2-nonenal (HNE), and isoprostane levels are good indicators of cellular redox stress [94], while HNE also promotes antioxidant response signalling [100,101]. Cholesterol also reacts with RONS to form cytotoxic oxidation products like 7α,β-hydroxy-,

$7\alpha$-hydroxyperoxy-, 7-oxo- and 5,6-epoxycholesterol, of which cholesterol modification by CP has been inferred [102].

Cancer cells have shown increased susceptibility to RONS (and hence CP) compared to non-malignant cells. This is because cancer cells have higher basal redox levels than normal cells due to increased metabolism, proliferation, dysfunctional mitochondrial activity, and low inducible antioxidant capacity which makes cancer cells more sensitive to redox stress [103]. Susceptibility to redox stress is further exacerbated by the elevated aquaporin overexpression [104], increasing $H_2O_2$ permeability and consequent lipid peroxidation [105], and lowering membrane cholesterol levels increasing permeability of ROS ($O_2^{\bullet-}$, $H_2O_2$) through cell membranes [106]. Furthermore, there is a strong inverse correlation between membrane cholesterol levels and sensitivity to CP [107], and cholesterol content can reduce membrane disruption by CP in a multilamellar liposomal model [99]. It is hoped that these factors together enable a goldilocks zone of CP treatment for simultaneous killing of cancer tissue while promoting healing and angiogenesis in healthy tissue.

In addition to directly reacting with membrane lipids and causing antioxidant (glutathione; GSH) depletion, $^1O_2$ (from CP and secondary decomposition of long-lived RONS in PAL) can deactivate catalase, amplifying $H_2O_2$ accumulation via NADPH oxidase (NOX)/superoxide dismutase (SOD) and mitochondria-dependent apoptosis via caspase 9/3 [108]. Cell metabolism (NADPH reduction) correlates with a higher tolerance to CP treatment [107]. Other indirect evidence supports this claim. For example, pyruvate protects cancer cells in PAM in vitro by scavenging $H_2O_2$ [109]. Pyruvate is an essential carbon source for cancer cell metabolism, and is in high concentration in some malignant tumour cells due to the "Warburg effect" driving preferential metabolism towards the lactate pathway [110]. This study suggests that cellular metabolism confers protection to cancer cells against CP, but more studies are required to investigate the effect of metabolic pathways in cancerous and non-malignant cells.

CP-derived RONS also readily react with protein side-chains, particularly aromatic groups of tryptophan, tyrosine and histidine, amine groups in lysine and arginine, cysteine thiol (-SH), and methionine sulfhydryl groups [111]. $H_2O_2$ oxidises several types of protein residues as a redox switch to signal gene up/downregulation, proliferation, migration, and metabolism [112]. Additionally, the lipid oxidation products mentioned earlier can also oxidise protein cysteine residues, resulting in lipid–protein adduct formation [113]. $ONOO^-$ can also spontaneously decompose into potent $^\bullet OH$ and $NO_2^\bullet$ radicals that react with protein tyrosine residues to form tyrosine radicals, inactivating proteins by forming irreversible 3-nitrotyrosine or dityrosine (protein–protein) adducts, or propagating lipid peroxidation [114]. $ONOO^-$ in CP and PAM is a potent driver of caspase-mediated apoptosis in malignant cells [108,115–117]. This could also occur in healthy cells with "higher" CP exposure, if it overcomes its antioxidant capacity.

GSH is the most abundant cellular antioxidant. The thiol moiety makes GSH an effective scavenger of RONS and oxidised proteins through *S*-glutathionylation (protein thiol–GSH adduct formation) to protect cells against redox stress [118]. In fact, cysteine supplementation before exposure to CP nearly completely negated biological responses and prevented GSH disulfide (GSSH) formation in skin keratinocytes (HaCaT) [119]. On the other hand, these protein modifications in excess led to protein inactivation by denaturation, cleavage, and aggregation, and can cause endoplasmic reticulum stress (ERS), initiating the unfolded protein response (UPR) [120]. Lipid (hydro)peroxides can also adduct to genomic DNA, impairing replication and transcription, leading to cell apoptosis [121]. Hence, these secondary byproducts of CP are advantageous in treating cancer, but need to be avoided in healthy cells. For an in-depth reading of CP-derived RONS on cell protein and lipid modifications, fantastic comprehensive reviews are available [21,94,111].

DNA is highly reactive to $^\bullet OH$ and $ONOO^-$, leading to genotoxic nucleoside -OH adduct radicals and oxidation products, sugar oxidation causing strand breaks and DNA-protein crosslinking [122]. Cells exposed to CP experience DNA oxidation (8-oxoguanine; 8-oxoG, or 8-hydroxy-2′-deoxyguanosine; 8-OHdG) due to ROS and attempt to repair damage

by expressing 8-oxoG DNA glycosylase (OGG1), poly(ADP-ribose) polymerase (PARP), and histone H2AX ($\gamma$H2AX) [123–125]. 8-oxoG was also formed in murine xenograft tumours treated with CP, accompanied by PARP activating p53/caspase 3-mediated apoptosis in vivo [125]. Furthermore, CP can have an epigenetic effect on cells, as shown in a breast cancer cell line that widely alters gene methylation, including inhibiting heat shock cognate B (*HSCB*) and phosphoribosyl pyrophosphate synthetase 1 (*PRPS1*) oncogene expression [126]. Obviously, genotoxic effects of CP would be harmful and possibly mutagenic if inflicted on healthy tissue. Fortunately, no elevated mutation rate due to CP exposure has been observed in HaCaT cells, fibroblasts, and lymphocytes in vitro using various CP devices [127–130]. Furthermore, no neoplastic lesions, signs of tumour growth, or tumour markers were found in 84 immunocompromised mice 350 days after receiving 14 consecutive daily CP treatments [131], nor DNA damage ($\gamma$H2AX) was observed in ex vivo CP-exposed human skin samples [132]. Although no tumour recurrence or adverse effects in proximal normal tissue were observed in a Phase I clinical study with 32 months of follow-up [82], long-term safety of CP devices regarding mutagenicity is still needed.

## 4. Cellular Responses to Cold Plasma through Redox-Responsive Intracellular Signalling Pathways

Redox signalling and homeostasis are ubiquitous to all life on Earth, involving a complex and dynamic cell-dependent system that acts as both sensor and effector of cellular environmental changes to coordinate response to stimuli (including during tissue regeneration, wound infection, or cancer). CP and PAL are potent sources of RONS that directly affect cellular redox homeostasis. However, redox responses can also be secondary, tertiary, etc., to the initial CP exposure [2,21]. Essentially, all therapeutic applications of CP are classified into the field of applied redox biology [25]. This has been exemplified by the scavenging of CP/PAL-derived RONS with the antioxidant N-acetylcysteine (NAC) that unanimously reduced or ablated DNA damage, cell cycle arrest, ERS, autophagy, and apoptosis to CP/PAL in several in vitro studies in cancer and prevented wound healing and antioxidant response in normal cells [116,123,124,133–153].

Cells can either be directly treated with CP while in media or indirectly treated with PAL, usually PAM or PAW. This leads to the accumulation of RONS, which dissolve from the plasma/gas phase into the liquid phase, or are generated by a complex series of secondary reactions (Figure 1) [29,154–156]. In cases of direct CP treatment of media containing cells, shorter-lived highly reactive oxidant species would also be pertinent in redox signalling. However, RONS such as NO$^\bullet$ and $^\bullet$OH are too reactive to diffuse further than a $\mu$m range in the liquid phase, but may form again through the decomposition of ONOO$^-$ [29,31,32]. Hence, the most commonly measured RONS are aqueous $H_2O_2$, $NO_2^-$, and $NO_3^-$ as markers of RONS generation, due to their long lifetime. Research to date has found that ultraviolet photons play a negligible role in eukaryotic cell responses to CP, with CP-derived UV dose being an order of magnitude below the minimal erythema dose and the UV not even being able to reach living skin cells [157]. However, CP-derived UV photons have also been shown to generate minor amounts of atomic H$^+$, $^\bullet$OH, and NO$^\bullet$ through photolysis of other RONS [35,158], which experiments with antioxidants cannot discriminate.

Epithelial cells are the most affected by CP, and illicit a host of molecular responses over time. Generally, oxidative stress leads to protein unfolding and cessation of protein synthesis (ERS/UPR), cell cycle arrest (G2/M phase), mitochondrial dysregulation, and post-translational modifications (predominantly phosphorylation via kinases) following the first few hours of CP exposure [159]. Cell cycle arrest and protein synthesis cessation/unfolding are instated quickly to preserve cell survival early after exposure, while DNA repair occurs later. The nuclear factor erythroid-2 related factor 2 (Nrf2) antioxidant response to oxidative stress is the most prominent cell response occurring immediately following, and up to several days after CP exposure to condition cells against redox stress, eventually restoring cell function and proliferation [159]. In this way, light CP exposure can

promote faster healing through redox eustress, levels of redox imbalance that condition cells to altered redox (usually oxidative) state, while returning to redox homeostasis. CP can also elicit autophagy, the cellular process of breaking down damaged components and organelles in the lysosome [160], and eventually apoptosis in cancer cells by amplifying redox stress [109,117,149,150]. Several distinct protein signalling pathways regulate these processes, and can cross-communicate to affect cell survival and proliferation [161]. For example, CP-activated extracellular signal-regulated kinase 1/2 (ERK1/2) mitogen-activated protein kinases (MAPK) and Akt, the latter leading to significant activation of nuclear factor-kappa B (NFκB) phosphorylation to promote cell proliferation [162]. Interestingly, nearly the opposite has occurred in cancer cells. CP caused apoptosis through simultaneously inhibiting Akt/mammalian target of rapamycin (mTOR) while exciting Jun N-terminal kinase (JNK) and p38 MAPK signalling pathways, leading to autophagy and the activation of p53-mediated activation of proapoptotic caspases, while inactivating Akt also inhibited NFκB to reduce tumour cell proliferation [163,164]. CP application in cancer therapy commonly leads to cell cycle arrest among different cancer types, and eventually autophagy and apoptosis [165]. These two examples demonstrate the hormesis approach to killing malignant cells while also promoting the healing of healthy tissue. This Section will discuss modulations of several cell signalling pathways affected by CP and PAL in the context of wound healing and cancer.

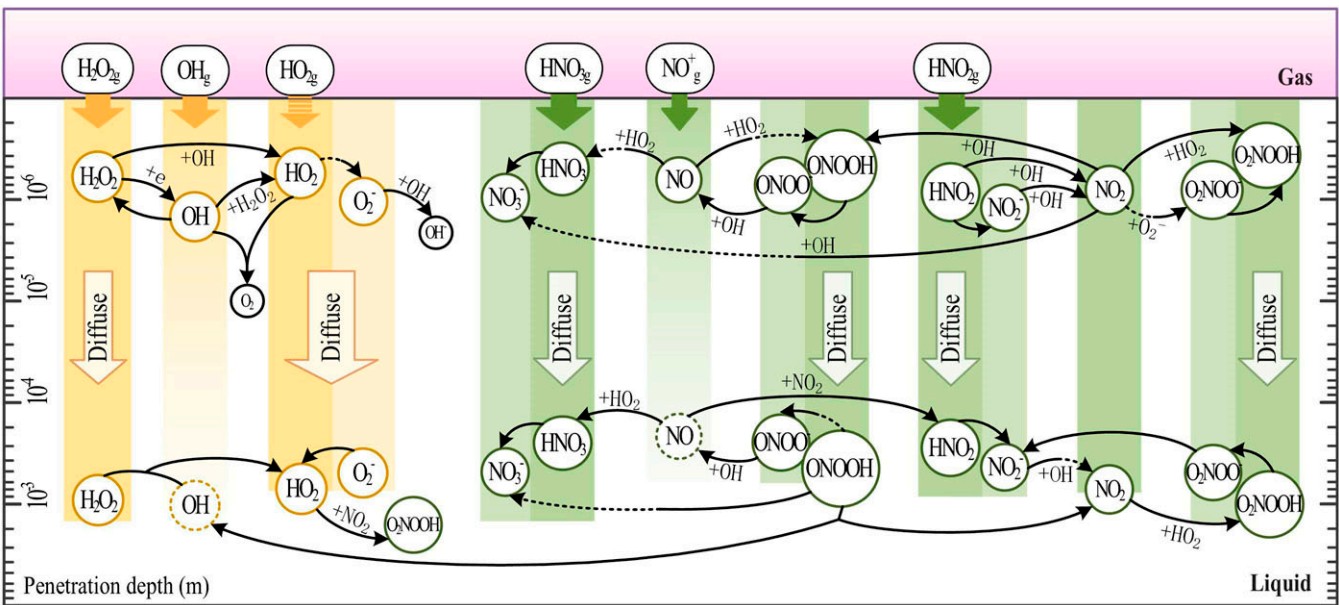

**Figure 1.** Chemical reactions of cold plasma (CP)-derived reactive oxygen and nitrogen species (RONS) at the gas–liquid interface and through diffusion in the liquid phase. The depths at which reactions occur are indicated on the vertical axis, with important and negligible reactions denoted by solid and dashed lines, respectively. Large orange (ROS) and green (RNS) vertical bars with "diffuse" arrows indicate diffusion as the dominant factor in the movement of RONS in the depth of liquid. Redistributed from [29] (**CC BY 4.0**).

*4.1. Keap1-Nrf2-Antioxidant Response Pathways*

Both cancerous and non-cancerous cells are armed with antioxidant response mechanisms to maintain redox homeostasis and affect survival, proliferation, and migration [159]. In fact, proteomic analysis revealed that the response to oxidative stress by Nrf2 is the most prominent cellular response to CP exposure in epithelial tissues [159]. The Nrf2 pathway acts as a redox-sensitive activator of the antioxidant response element (ARE). Under normal cellular homeostasis, Nrf2 rests in the cytosol bound to Kelch-like ECH-associated protein 1 (Keap1), keeping Nrf2 inactive and enabling E3 ubiquitin ligase-dependent proteasomal degradation [166]. Keap1 has canonically acted as the redox stress sensor, as thiols in

multiple Keap1 cysteine residues are sensitive to CP-derived RONS [167]. When these cysteines are oxidised, Nrf2 is released, translocates to the nucleus, ligates to small musculoaponeurotic fibrosarcoma (sMAF) proteins on DNA and transcribes ARE genes [166]. The ARE encodes a host of antioxidant proteins and enzymes to scavenge RONS that may harm vital cellular components, including GSH, GSH metabolism-related enzymes GSH reductase (GR), GSH peroxidase (Gpx), GSH S-transferase (GST), $\gamma$-glutamylcysteine ligase catalytic/modifier (Gclc/m) subunits, antioxidant enzymes SOD, catalase, haem oxygenase 1 (HO1), thioredoxin (Trx), thioredoxin reductase (TrxR), sulfiredoxin (Srx), and multifunctional stress response enzymes UDP-glucuronyl transferase (UGT) and NADPH quinine oxidoreductase 1 (NQO1) until redox homeostasis is returned [168].

Clear evidence shows that redox signalling is essential to wound healing [169]. Chronic wounds occur in parallel with chronic redox stress, which amplifies inflammation, apoptosis, and impedes neovascularisation [169]. As a consequence, therapeutic treatments that modulate redox status in interstitial tissue and cells are coming into focus to remedy chronic wounds [169], including CP induction of antioxidant response mechanisms via Nrf2 signalling (Figure 2). When exposed to CP, keratinocytes (HaCaT) increased HO1, catalase, NQO1, GST, and Gclc/m transcription via Nrf2 activation and translocation, concurrent with the release of proangiogenic chemokine VEGFA, growth factors heparin-binding epithelial growth factor (HBEGF), colony-stimulating factor 2 (CSF2), Prostaglandin-endoperoxide synthase 2 (PTGS2), and inflammatory cell cytokine interleukin 6 (IL6), which promote re-epithelialisation and wound repair [170–173]. While Nrf2 disassociates from Keap1, Keap1 also alters cytoskeletal arrangement, regulating E-cadherin and F-actin filamentation and critical during tissue regeneration [174,175]. In conjunction with reducing connexin 43 (Cx43) expression to relax cell–cell focal adhesion in skin cells [174], CP increases keratinocyte motility to re-epithelialize wounds. The role of Keap1 in response to CP has so far been limited to its role as a redox switch for Nrf2 and as a cytoskeletal junction protein. Yet, Keap1 is involved in several regulatory roles in the cell ranging from protein degradation, promoting NF$\kappa$B p65 translocation for cell survival, cell cycle progression, and p62-mediated autophagy [176]. Nevertheless, there is growing momentum towards studying Keap1 beyond its canonical function as an inhibitor of Nrf2, but this is yet to be explored with regard to CP treatment.

When translated into mice studies, direct CP treatment of acute full-thickness wounds accelerated wound healing, neutrophil, and macrophage infiltration and TNF$\alpha$, TGF$\beta$ and IL-1$\beta$ transcription in vivo, and promoted HO1 and NQO1 transcription in dermal fibroblasts and epidermal keratinocytes via Nrf2 ex vivo [174,177]. THP1 monocytes also activate Nrf2 and upregulate HO1 transcription in response to CP exposure [178,179]. Studies to date have also shown that RONS, like ONOO$^-$, can also activate Nrf2 indirectly via the PI3K/Akt pathway [180,181], and upon overexposing keratinocytes to CP at levels that significantly damage lipids, proteins, and DNA, Nrf2 is suppressed as a result of Akt degradation [124]. Similarly, inhibiting JNK (but not ERK or p38) attenuated CP-mediated HO-1 induction [138], indicating that CP-derived RONS may cross activate Nrf2 through JNK, although the mechanism for this effect is yet to be determined.

HO1, an integral product of Nrf2, catabolises haem into biliverdin, releasing Fe$^{2+}$ and carbon monoxide (CO), with biliverdin being further degraded to bilirubin by biliverdin reductase [182]. Bilirubin, CO, and ferritin upregulation by Fe$^{2+}$ protect cells through RONS scavenging to inhibit apoptosis and inflammation [183]. This in, concert with increased GSH synthesis, seems to support Akt activation, promoting cell survival by inducing Nrf2/ARE and inhibiting p53 and Bax/Bcl2-dependent caspase-mediated apoptosis. Moreover, p53 performs a biphasic function in Nrf2 modulation; low p53 activity enhances Nrf2 expression to foster cell survival, whereas high p53 activity suppresses Nrf2 to promote cell death [184]. In this sense, the p53/Nrf2 axis acts as a focal point in deciding the fate of the cell, and may be a mechanism by which CP can be applied therapeutically on a hormesis basis.

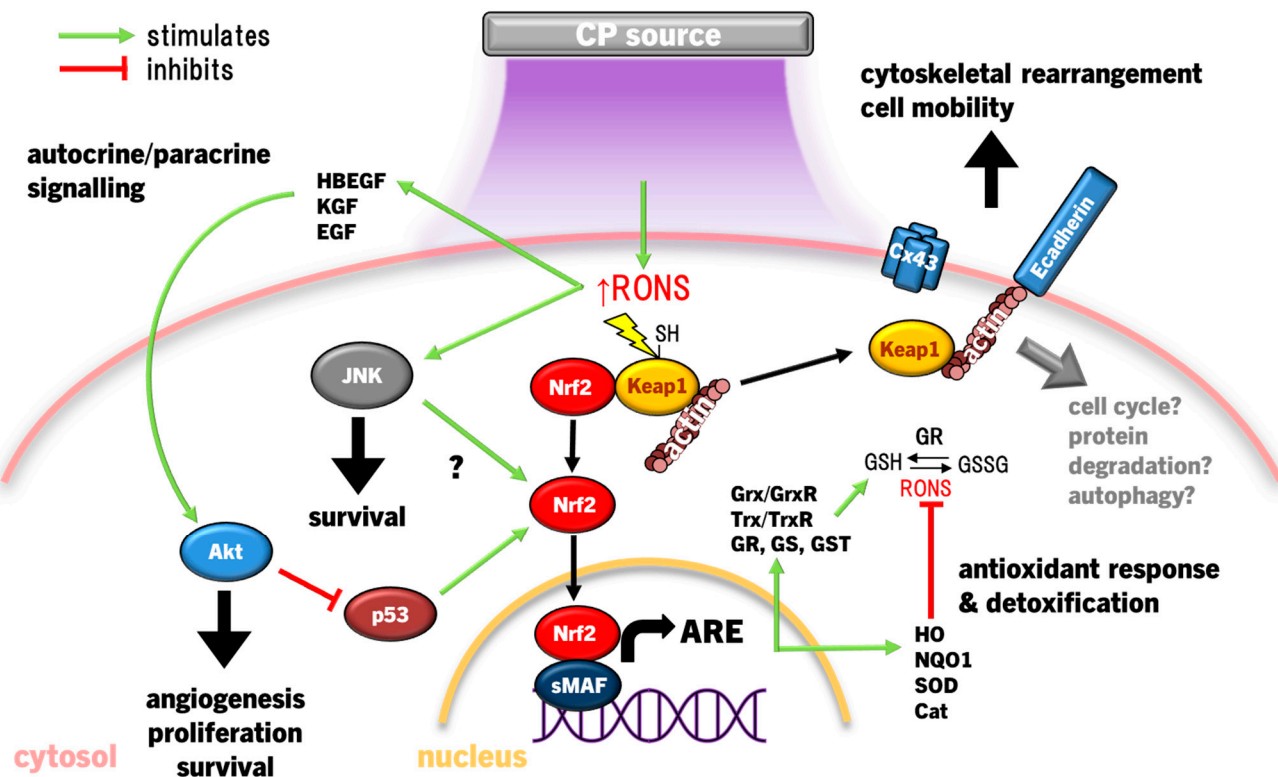

**Figure 2.** CP-derived RONS promote wound healing by dissociating Kelch-like ECH-associated protein 1 (Keap1)/nuclear factor erythroid-2 related factor 2 (Nrf2) to activate the antioxidant response element (ARE), rearranging cytoskeletal architecture to promote cell motility, recruiting inflammatory cells, and autocrine signalling to promote cell survival, proliferation, and angiogenesis. The Jun N-terminal kinase (JNK) pathway has been shown to potentiate Nrf2 activation, but the mechanisms are still unclear (indicated by "?"), while Akt decreases p53 expression to promote Nrf2. Keap1 activity may also go beyond the canonical function of inhibiting Nrf2, but these functions are greyed out to indicate the unexplored nature of these potential responses to CP.

In many cancer cells, *Keap1* and *Nrf2* gene mutations constitutively increase their activity [185], acting as counterbalance to promote survival in the higher endogenous RONS environment resulting from malignant metabolic and proliferative activity [186]. Inhibiting HO1 activity or *Nrf2/HO1* gene silencing results in significantly greater apoptosis by CP [138]. Therefore, Nrf2/HO1 inhibition and CP adjunct therapy could synergise as a cancer therapy, similar to HO1 inhibition improving conventional chemo/radiotherapy and reducing tumour growth [187], or synergistic activity of CP with chemo/radiotherapy in vitro and in vivo [139,145,188–194]. Unfortunately, the absence of clinically safe HO1 inhibitors currently precludes testing, but may be grounds for future clinical research. Simultaneous treatment with CP and pharmacological inhibition of Trx also synergistically enhanced caspase 3-dependent apoptosis with some degree of lipid peroxidation-dependent ferroptosis (iron-dependent form of apoptosis linked to cytotoxic lipid peroxide accumulation) in glioblastoma cancer cells and slowed tumour growth in glioblastoma-bearing mice [145]. As glioblastomas are incurable, manifest as the most aggressive and deadly tumours of the brain, and are rapidly growing in incidence [195], CP therapy could have a massive impact on improving survival and prognosis of these patients.

Recently, it was observed that CP activates Nrf2 concurrent with BTB and CNC homolog 1 (Bach1) in THP1 cells [179], an antagonist of Nrf2 that competitively binds with sMAF proteins as a negative-feedback to ARE [196]. This is in contrast to previous work showing that CP enhances Nrf2 and suppresses Bach1 transcription in HaCaT [170]. The reason for these contrasting cellular responses remains unclear. Considering that Bach1

is degraded in cells under oxidative stress [197], Nrf2 activity may paradoxically lead to upregulation of Bach1 in cancer cells [198], and as Bach1 is an important oncogene that drives tumour metastasis [198,199], comparing the response of Bach1 to CP in normal and cancer cells should be investigated. Overall, Nrf2-ARE activation following CP treatment is integral to restoring redox homeostasis and cell survival with the interplay with p53, JNK, and Akt. Additionally, Nrf2 may only be half the story, as the canonical Nrf2 inhibitor, Keap1, also signals cytoskeletal protein rearrangement and cell junction proteins to promote wound healing. However, an increasing body of research shows that Keap1 also influences cell fate decisions [176], which should be further investigated as it relates to CP.

### 4.2. ERS and UPR Signalling Pathway

Under redox imbalance, cells experience endoplasmic reticulum (ER) stress (ERS), causing elevated protein misfolding that impairs cell function. In an attempt to counteract this, the ER releases calcium to signal general suppression of protein synthesis, while upregulating a suite of chaperones that repair protein misfolding as the unfolded protein response (UPR) to restore homeostasis [120]. There are three arms of the UPR that signal cells to increase their protein folding capability: (i) the inositol-requiring protein $1\alpha$ (IRE1$\alpha$), which uniquely splices X-box binding protein 1 (Xbp1) mRNA to express the Xbp1 transcription factor, (ii) protein kinase RNA-like ER kinase (PERK) that activates the eukaryotic initiation factor 2-alpha (eIF2$\alpha$), and (iii) activating transcription factor 6 (ATF6), which gets cleaved in the Golgi apparatus to translocate to the nucleus [120]. All three arms promote chaperones and recovery of ER biosynthesis activity to restore protein synthesis, while IRE1$\alpha$ and ATF6 also signal inflammatory responses. However, only prolonged PERK activation can decide cell fate via C/EBP-homologous protein (CHOP) in two ways. Firstly, if the ER recovers, CHOP acts as negative feedback to PERK activity by promoting GADD34 and constitutive repressor of eIF2$\alpha$ phosphorylation (CReP) binding to protein phosphatase 1 (PP1) which dephosphorylates (deactivates) eIF2$\alpha$ [200,201], or if ERS go unresolved, prolonged CHOP activation activates apoptosis [202]. One of the integral ERS-associated proteins that signals ERS and initiation of the UPR is glucose-regulated protein 78 (GRP78). GRP78 is expressed on the ER membrane as an ERS sensor bound to PERK, IRE1$\alpha$, and ATF6, and acts as a chaperone that binds to misfolded proteins [203]. Therefore, the UPR can be enacted by both normal and malignant cells in an attempt to restore cellular homeostasis, including in response to redox dysregulation.

Later, in airway epithelial cells, induction of the UPR in response to CP was confirmed, revealing significant upregulation of a number of ERS-associated proteins, including GRP78 [159]. Importantly, apoptosis was absent in conditions of lower CP exposure times in these studies [159,171]. While these studies have shown that CP can induce ERS/UPR, CP has also been shown to remediate cells already experiencing ERS. In a chemically induced atopic dermatitis model in vitro and in vivo, elevated inflammation (upregulated TNF$\alpha$, IL1$\beta$, and C-C motif ligand 2 (CCL2), with decreased anti-inflammatory IL10), ERS/UPR, GRP78 expression, and eventually CHOP-mediated apoptosis were observed, all of which were curtailed by CP treatment via induction of HO1 [204]. HO1 induction as part of the response to CP indicates the role of Nrf2 in conditioning the cell for restoring redox homeostasis to cellular function [177]. Altogether, these results show that inducing moderate ERS with CP may activate the UPR in a pro-survival response to redox eustress, while other cell signalling pathways promote wound healing (Figure 3).

ChaC GSH-specific $\gamma$-glutamyl cyclotransferase 1 (CHAC1) degrades GSH, and thus plays a role in redox balance of the cell [205] and is also overexpressed by prolonged activation of the PERK/IRE1$\alpha$ arm of the UPR [206]. CHAC1 expression was not induced in keratinocytes treated with PAM activated for only 20 s with CP, but was transcriptionally induced up to 5-fold with PAM activated for 180 s [171]. Therefore, the lack of cell death and no CHAC1 induction with lower CP activation time showed PAM to be safe on skin cells [171]. On the other hand, while the ERS/UPR may be exploited with CP to promote survival in healthy cells, prolonged redox dysregulation in cancer cells may also be possible

through aberrant activation of the UPR. The expression of CHAC1 in breast and ovarian cancer cells and associated with significantly higher mortality in breast and ovarian cancer patients, implicating a possible role as a metastatic factor in these cancers [207,208]. Furthermore, *CHAC1* was among the most upregulated genes in SCC and glioblastoma cells (>16 and 25-fold, respectively) following treatment with PAM [209,210] that was sizably larger than in keratinocytes [171], indicating an intense degree of redox stress aggravating the UPR in cancer cells that led to apoptosis [209]. In summary, while slightly elevated CHAC1 expression may increase malignancy of cancer cells [207,208], overexpressing CHAC1 with CP may kill cancer cells via prolonged PERK/IRE1α activation [209]. Activation of the PERK/eIF2α and IRE1α/Xbp1 arms of the UPR, upregulation of GRP78, and eventually CHOP expression leading to apoptosis have also been observed in colorectal and melanoma cancer cells exposed to CP [151,211]. Neuroblastoma cells in response to CP exposure also activated eIF2α that led to stress granule formation (protein-RNA complexes that interrupt mRNA translation), in line with PERK-mediated UPR [212]. To date, ATF6 activation has not shown involvement in CP-induced UPR, and was not activated in melanoma cells [211]. The ERS/UPR is predominantly a result of elevated redox stress caused by CP, as NAC was able to prevent the UPR [151]. Unfortunately, all evidence of CP-inducing ERS and UPR to date are in vitro, while only one study showed CP could remediate already present ERS and UPR in a mouse atopic dermatitis-like in vivo model [204]. The relevance of the UPR has not yet been confirmed in vivo with CP treatment of normal tissue, wounds, or malignant tissue.

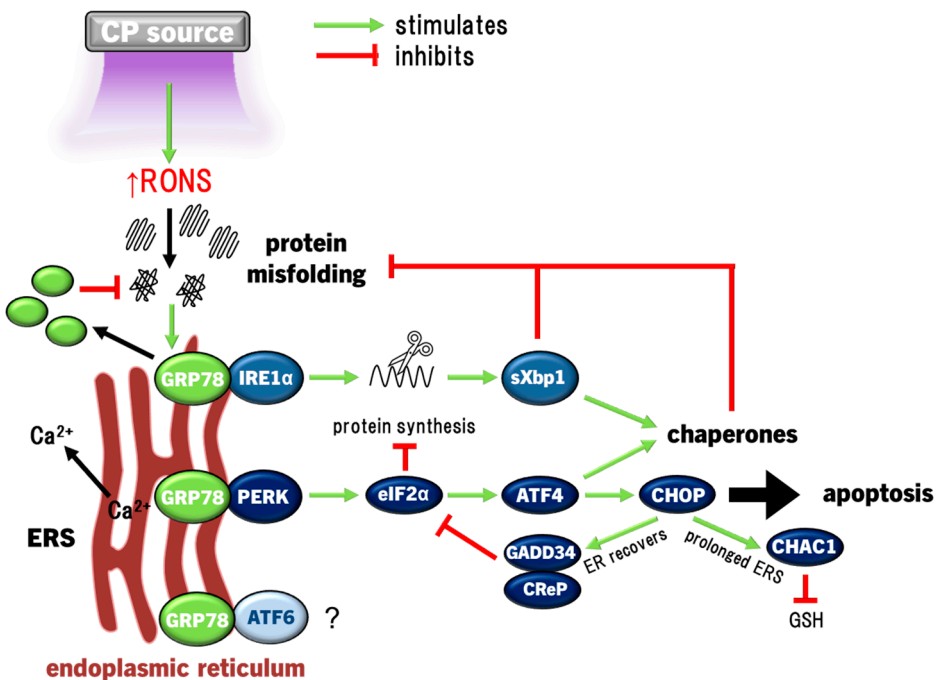

**Figure 3.** CP exposure causes endoplasmic reticulum stress (ERS) due to redox stress and activates the inositol-requiring protein 1α (IRE1α) and protein kinase RNA-like ER kinase (PERK) arms of the UPR. Activating transcription factor 6 (ATF6) activity has not been observed (indicated by "?"). Lower CP exposure allows cells to recover, while higher CP exposure can cause prolonged C/EBP-homologous protein (CHOP) activity, leading to apoptosis.

### 4.3. ERK1/2, JNK, and P38 MAPK Pathways

The mitogen-activated protein kinases (MAPK) are a family of serine/threonine kinases categorised into three canonical signalling cascades: the extracellular signal-regulated kinases (ERK1/2), c-Jun N-terminal kinase (JNK), and p38 MAPK. The role of MAPK in cells is complex and influences diverse cellular functions. The ERK1/2 signalling cascade generally promotes pro-survival actions like regulating the cell cycle in response to stress,

but also proliferation [213], whereas the JNK and p38 pathways usually balance cell fate decisions regarding autophagy and apoptosis [214]. However, the long-term induction of these pathways also leads to terminal cell death. For instance, JNK and ERK1/2 can both propagate caspase-mediated apoptosis and autophagy, while JNK and p38 activate p53 to enact intrinsic (mitochondria-mediated) and extrinsic (Fas death receptor-mediated) apoptosis [24]. Therefore, the complex dynamics of MAPK activity allows these pathways to both promote survival and activation of healthy cells and kill malignant cells when applied through a hormesis approach.

4.3.1. MAPK Inducing Cancer Cell Death

MAPK signalling itself and through cross-communication with PI3K (another oncogenic protein) are some of the most prominent pathways causing cancer malignancy by promoting cancer cell survival and proliferation [185,215]. In particular, MAPK is frequently mutated in cancer cells, which leads to constitutive MAPK amplification from the nascent pre-malignant stage through to tumour progression and metastasis [216]. Accordingly, this makes modulating MAPK a promising prospect for anticancer therapy. As a result, MAPK signalling alterations in cancer cells and tumours has been well characterised, often showing to lead to impaired invasiveness, cell cycle arrest, and eventual cell death.

Exposure of cervical cancer cells (HeLa) to CP reduced ERK1/2 and JNK activity, while JNK and p38 activity was reduced in ovarian cancer cells exposed to PAM, leading to significantly diminished invasiveness (cell migration) and matrix metalloproteinase 9 (MMP9) activity [147,217]. However, more lethal exposures to CP markedly activate JNK and p38 in several cancer cell types, including HeLa cells, neuroblastoma, prostate cancer, lung carcinoma, hepatoma, breast cancer, and colorectal adenocarcinoma cell lines, leading to cell cycle arrest, then bcl2 proapoptotic proteins bax and bak initiate mitochondria-dependent cell death via cytochrome c release and caspase 9/3-driven apoptosis [39,116,123,142,164]. In several cancer cell types with high gasdermin E (GSDME) expression, CP treatment also led to pyroptosis [142], a highly inflammatory manifestation of programmed necrosis caused by gasdermin (including GSDME) cleavage by caspase 3, leading to GSDMN pore formation that perforates the cell membrane [218–221]. ERK1/2 expression and phosphorylation in cancer cells can also be significantly downregulated by CP [144,222,223]. Exposure of thyroid papillary cancer cells to CP led to a dismantled cytoskeletal orientation and decreased MMP2/9 activity that impaired cell invasiveness, which was not observed in normal thyroid cells [222]. In addition to inactivating pro-apoptotic factors, bcl2 also inactivates beclin 1, an essential protein for coordinating the formation of the early autophagosome [224]. Concurrently, CP and PAL can also increase ERK1/2 activity in cancer cells, leading to LC3 activity, which, in coordination with JNK inhibiting bcl2 to release beclin 1 and upregulating autophagy-related protein 5 (ATG5), leads to the development of the nascent autophagosome [149,188,225]. CP also upregulates the expression of pro-autophagic sestrin 2 via JNK to initiate the extrinsic apoptosis pathway [148,226]. Autophagy is essentially a survival mechanism, whereby cells can survive if autophagy resolves the damage. However, cancer cells being more susceptible to RONS causing damage tends to lead to caspase 3-driven apoptosis. Therefore, the susceptibility of malignant and healthy cells to RONS is also affected by MAPK signalling, as summarised in Figure 4.

Downregulation of ERK1/2 and upregulation of the JNK pathways due to CP/PAL are usually accompanied with decreased PI3K/Akt/mTOR pathway and downstream NFκB activity in cancer cells to inhibit proliferation and promote caspase 9/3-dependent apoptosis [144,163,210]. $ONOO^-$ has been implicated as a major causative of RONS due to the high detection of its biomarker 3-nitrotyrosine in affected cells [115]. The PI3K/Akt pathway is an inhibitor of p53 [227], which is antagonistic to the p53-stimulating effect of p38 and JNK MAPK pathways [228,229]. Taken together, CP induces apoptosis by the inverse regulations of PI3K/Akt, p38, and JNK, amplifying p53 expression and signalling. Furthermore, JNK activation of caspases 9/3, suppression of ERK1/2 and PI3K/Akt, and mitochondrial dysfunction are the result of the lack of antioxidant capacity

to repair damage caused by CP-derived RONS [108,230,231]. This is clearly evidenced by the frequent findings that MAPK activation, together with downstream cancer cell invasiveness, apoptosis/pyroptosis, and autophagy is ablated by the preconditioning of cells with the RONS scavenger NAC [116,123,142,144,145,147–149,152]. $H_2O_2$ may be the major culprit, given that selective $H_2O_2$ quenching also prevented cell cycle arrest and death [145,164], in addition to the increased sensitivity of cancer cells to $H_2O_2$ due to elevated aquaporin expression and lower membrane cholesterol content [104,107], as described earlier.

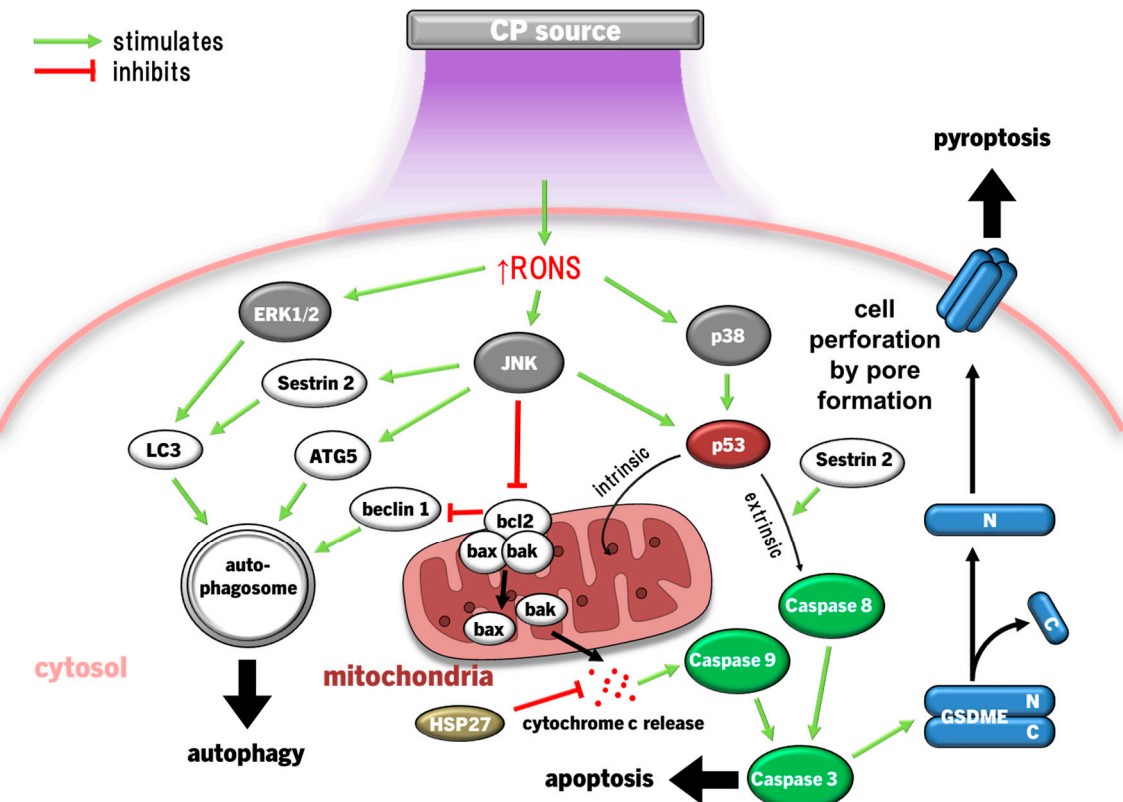

**Figure 4.** CP-derived RONS promote cancer cell autophagy and death by activating the mitogen-activated protein kinase (MAPK) family signalling pathways. RONS activating JNK and extracellular signal-related kinases 1/2 (ERK1/2) pathways leads to autophagosome formation and autophagy, while JNK and p38 also potentiate apoptosis via both intrinsic and extrinsic pathways of p53. Cancer cells that highly express gasdermin E (GSDME) also undergo pyroptosis.

Various preclinical mouse models of tumour progression have been performed to test the efficacy of CP as an antitumour therapy [147,152,223,232]. A reduced rate of glioblastoma and ovarian tumour progression was observed when the respective cancer cells were pretreated with PAM/CP prior to xenografting, with immunohistochemical analysis showing increased p38, JNK, and caspase 3 activation in CP-treated tumours [147,232]. However, pre-treating cancer cells with CP in vitro before xenografting lacks clinical relevance. Two other studies in BALB/c mice with lung cancer cell and glioblastoma cell xenograft tumours were treated with CP daily for 10 and 20 days, respectively [152,223]. Glioblastoma tumour growth was significantly decelerated with CP treatment, with immunohistological analysis showing extensive DNA damage, NOX3 expression (indicating endogenous $H_2O_2$ production), and caspase 3 activation, which implicate JNK and p38 activity, but tumour growth still persisted [152]. In contrast, 10 days of CP treatment trended towards reducing tumour progression, but this was not significant [223]. Regardless, both of these studies showed that CP induced JNK and p38 activity to slow tumour growth, but CP therapy alone was not effective enough to halt tumour progression. Fortunately, mouse xenografted

tumour models have shown that combination therapy of CP/PAL and chemotherapeutic drugs can synergistically reduce tumour growth rate and even lead to tumour size reduction [139,145,191,193,194]. However, more animal models that test adjunct CP treatment with chemotherapy against progressive tumours are warranted to determine parameters for CP-augmenting chemotherapies before progressing to clinical trials.

4.3.2. MAPK in Wound Healing and Inflammation

The hormesis principle applied to CP treatment of healthy cells to promote healing potentiates, a different profile of MAPK activity than that observed in destroying cancer cells. Exposure of keratinocytes to CP has led to increased gene expression of MAPK and MAPK kinases (MEK), with downstream expression and release of IL6, IL7, IL8, IL3 receptor (IL3R), IL4R, and IL6R in keratinocytes [171,233], which collectively promote the inflammatory response, leukocyte recruitment, monocyte activation, and differentiation into macrophages [234]. Growth factors, including granulocyte macrophage colony-stimulating factor (GMCSF), HBEGF, and VEGF, were also upregulated [171], and these factors have demonstrated a critical role of CP indirectly stimulating proliferation, migration, and activation of MAPK through autocrine actions of growth factors. Shorter CP exposure times (20 s) marginally activated MAPK, but cytotoxic exposures (180 s) markedly activated all arms of MAPK signalling, upregulating HSP27 to arrest the cell cycle and prolonging stimulation of p53 to induce apoptosis [233]. Upregulation of HSP27 likely acts as a negative feedback mechanism to delay apoptosis [235], as HSP27 is known to prevent cytochrome c-dependent activation of caspase 3 [236], but CP exposure in this experiment was too harsh to prevent apoptosis [233].

Similar results were also observed in THP1 monocytes in vitro that upregulated JNK and p38 phosphorylation, as well as HSP27 in response to PAM [237]. Interestingly, leukaemia (Jurkat) lymphocytes are far more sensitive to PAM, inducing significantly stronger JNK and p38 phosphorylation and caspase 3-mediated apoptosis with low HSP27 detection [237]. For isolated human monocytes, PAM formed from 30 to 60 s CP exposure stimulated monocytes through MEK-ERK1/2 [238], concurrent with elevated proinflammatory IL8 release [239]. Conversely, 3–6 min CP exposure induced phosphorylation of ERK1/2 and JNK pathways, leading to caspase 3 activation, finding that JNK activated $H_2O_2$ concentration-dependent in PAM [238]. Therefore, data to date suggests that low CP exposure time activates monocytes and has been shown to promote the proinflammatory M1 (antitumour) differentiation of macrophages [240]. Conversely, high CP exposure times may kill immune cells that may help modulate inflammation, which would not be beneficial in chronically inflamed wounds.

Compared to the antitumour testing of CP and PAL in animal models, there is a paucity of animal experiments translating CP-induced effects of MAPK in wound healing. What has been found is that CP and PAW treatment significantly accelerated wound healing in Sprague Dawley rats with full thickness excisional wounds, culminating in >90% wound closure approximately two to four days earlier than untreated wounds [153,241], with PAW-treated wounds reaching full wound closure 7 days faster than untreated wounds [242]. The CP-treated wounds exhibited significantly greater ERK phosphorylation promoting proliferation and N-cadherin expression with lower E-cadherin expression [241], which are crucial steps for detaching cell–cell junctions to promote cell migration to re-epithelialise wounds [243]. This is reciprocated by findings that PAW increased integrin β1/5 expression and phosphorylation of FAK and paxillin in keratinocytes, indicating that PAW promotes keratinocyte migration and proliferation in vitro and in vivo [153].

Neutrophils are an integral first responder in host defence against pathogenic microbes, but have received little attention for their response to CP. One study found that direct CP treatment of isolated neutrophils elicited the release of neutrophil extracellular traps (NET) and $NOX/O_2^{\bullet -}$-dependent NETosis [244]. NETs are the release of histone-conjugated genomic DNA bound to proteases into the extracellular space from neutrophils to trap microbes, and leads to NETosis, a form of programmed cell death specialised by

neutrophils [245]. Like CP treatment, NET formation has been linked to the activation of p38 and ERK1/2 MAPK [246]. The role of CP-induced NETs/NETosis in enhancing host defence is likely negligible, as CP itself is a broad-spectrum antimicrobial [40,41]. Then again, in chronic wounds which have high neutrophil abundance and dysregulated inflammation, CP may induce pathological NET release in the wound that can antagonise other wound healing processes by exacerbating prolonged inflammation [247]. However, this is still controversial, and further mechanistic studies are needed to validate this finding. Currently, the clinical relevance of NETs/NETosis on wound healing, due to CP or not, is still also unclear. Additionally, NETosis has been linked to increased tumourigenesis, and inhibitors of NET release are being trialled in conjunction with other cancer therapies (reviewed [248]), but NET formation in cancer as a result of CP treatment has not been investigated. Therefore, the possibility of NETosis due to CP treatment of tumours and cancer treatment should be investigated in future animal and clinical trials.

### 4.4. PI3K/Akt Pathway

The phosphoinositide 3-kinase (PI3K)/protein kinase B (Akt) signalling cascade orchestrates diverse regulatory functions in cell survival, disease pathogenesis, angiogenesis, and tumorigenic processes, and therefore it is an important pathway for both induction of wound healing and cancer intervention [216,249]. Healing of acute wounds normally occurs with basally elevated Akt/mTOR activity to promote cell growth, migration, and angiogenesis to help revascularise the new tissue [250,251]. In contrast, chronic wounds, including diabetic foot ulcers (DFU), have been shown to exhibit impaired PI3K/Akt/mTOR signalling that impairs cell survival and reduces growth factor release, which would also stimulate surrounding tissue to heal the wound [252].

PAM has demonstrably promoted keratinocyte proliferation in vitro and in vivo via promoting Wnt/β-catenin signalling and PI3K/Akt/mTOR [143,253,254]. Akt inhibits glycogen synthase kinase 3β (GSK3β), which prevents β-catenin degradation and allows nuclear translocation to promote cyclin D expression and the consequent G1/S proliferative cell cycle phase in keratinocytes [254]. Evidently, a hormesis approach to CP applies here, as the higher CP-derived RONS ($H_2O_2$ and possibly NO) exposures then reduced PI3K/Akt, ERK1/2, and β-catenin signalling back to untreated levels [143]. Another axis is shown whereby CP-induced PI3K/Akt activation leads to downstream ERK1/2 and NFκB activation to also promote cell survival and proliferation, likely involving CP-derived NO [143,162,253]. PI3K/Akt signalling also leads to NO synthase activation, which increases endogenous NO production [255], followed by activation of protein kinase G and downstream ERK1/2 to promote proliferation [256]. Therefore, not just CP-derived NO, but the stimulation of endogenous sources of NO are involved in promoting survival, proliferation, and angiogenesis, which are critical to wound healing.

Akt can also dampen pro-apoptotic p53 and Bax expression, leading to the upregulation of anti-apoptotic Bcl2 and increased Nrf2 activity to improve cell survival [177]. Skin-derived mesothelial cells exposed to PAM also had higher Akt expression and phosphorylation with dampened p53 activity to promote survival, while fibroblasts died through p53-dependent apoptosis [257]. Furthermore, CP treatment of skin cells also results in the release of epidermal growth factor (EGF) and keratinocyte growth factor (KGF), which promote autocrine skin cell proliferation, elevate MMP2/9 activity for ECM remodelling, and release VEGF to recruit endothelial cells to aid in angiogenesis [258]. Stimulation of PI3K/Akt by CP is summarised in Figure 5.

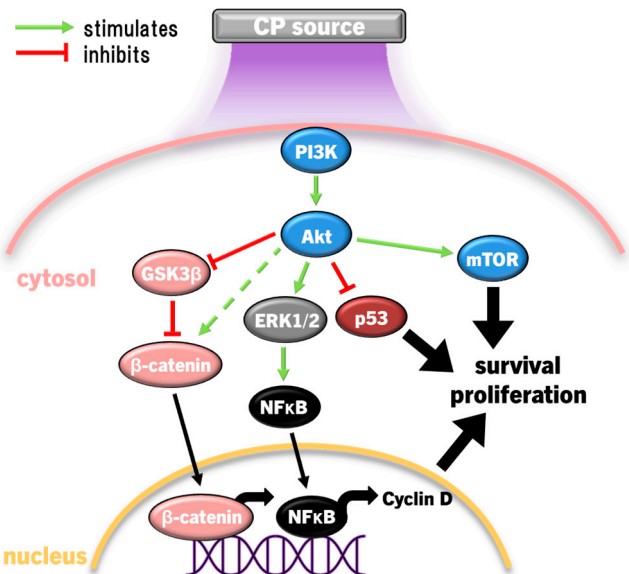

**Figure 5.** Phosphoinositide 3-kinase (PI3K)/Akt/mammalian target of rapamycin (mTOR) pathway stimulation by CP leads to multifactorial stimulation of cell survival and proliferation by indirectly stimulating the β-catenin pathway (indicated by dashed green arrow), activating nuclear factor-kappa B (NFκB) via ERK1/2 and inhibiting p53.

Suppressing PI3K/Akt Signalling in Cancer Cells

The PI3K/Akt signalling pathway is commonly overexpressed through genetic alterations in cancer cells leading to metastases [249]. In normal cells, active phosphatase and tensin homolog (PTEN) inhibits PI3K/Akt activation, whereas many types of cancers are PTEN-deficient, leading to aberrant PI3K/Akt signalling [249,259]. The use of CP in the destruction of cancer cells has been shown to occur with increased Akt degradation due to redox stress [109,117,144,149,150,191]. The first study in head and neck cancer (human SCC15 and mouse SCC7) cells found that CP increased apoptosis, concurrent with reduced Akt phosphorylation and expression through MUL1-encoded E3 ligase-mediated ubiquitination [150]. Furthermore, tumour growth in a mouse model of head and neck cancer was slowed and eventually stopped with CP treatment, with increased MUL1 expression and decreased Akt phosphorylation confirmed immunohistochemically [150]. MUL1 expression can also stimulate NFκB activity to inhibit apoptosis, and yet this is interpreted to be a short-term effect in which cells eventually succumb to apoptosis during prolonged stress [260]. Reduced Akt/mTOR expression has also been linked to reduced hypoxia-inducible factor 1α (HIF1α) expression to dampen proliferation [117]. Despite this, direct CP treatment of glioblastoma cells attenuated proliferation and decreased total Akt protein expression, but increased Akt and ERK1/2 phosphorylation 24 h after exposure [261]. Although Akt degradation occurred in both situations, differences in the activation of Akt observed between different cell lines in these studies may be due to the *PTEN* status of cancer cells. The U-87 glioblastoma cell line that elevated Akt phosphorylation in response to CP is a PTEN-mutant, while the LN-18 cell line is a *PTEN* wild-type and did not experience elevated Akt phosphorylation [261]. In view of this, the gene status of *PTEN* (among other tumour suppressor-related genes) may be a factor that affects CP therapy against cancers.

In addition to targeting PI3K/Akt in cancer, the signal transducer and activator of the transcription 3 (STAT3) pathway counterbalances PI3K/Akt/mTOR signalling via PTEN and promotes tumourigenic proliferation, invasion, migration, and angiogenesis in most cancers [262]. When Akt is inhibited in PTEN-deficient cancer cells, there can be a large increase in STAT3 signalling that compensates [263]. Therefore, anticancer therapeutic strategies that inhibit both PI3K/Akt and STAT3, particularly in PTEN-deficient cancers, could be more effective in halting tumorigenesis [263]. CP and PAM at lethal exposures also led to the deactivation of STAT3, which, together with the degradation of Akt signalling,

halted tumour growth and promoted caspase 3-driven apoptosis in osteosarcoma and pancreatic cancer cells [109,149,191].

PAM has also demonstrated efficacy towards destroying cancer cells [140,264]. PAM-treated human hepatoma (HepG2) cells had significantly decreased expression of Akt and mTOR, which led to decreased p62 (inhibitor of autophagy) and increased Beclin 1 and LC3-II expression to promote autophagosome formation [264]. The autophagosome formation, autophagy, and degradation of Akt/mTOR were all ablated by the simultaneous addition of catalase and SOD, implicating extracellular $O_2^{\bullet-}$/$H_2O_2$ as the causative PAM-derived ROS [264]. PAM also induced the expression of PTEN, and consequently reduced Akt phosphorylation and deactivated NFκB. However, it also sensitised cancer cells to the TNF-related apoptosis-inducing ligand (TRAIL) and greatly enhanced caspase 8/3-driven apoptosis [140]. Mechanistically, the increased PTEN activity was likely due to reduced the expression of miRNA425 that inhibits PTEN translation [140,265]. TRAIL binds to death receptors to induce apoptosis and is highly selective towards cancer cells, but aberrant PI3K/Akt pathway activity, common in cancer cells [249], can also lead to TRAIL resistance [266]. The importance of PTEN is accentuated by findings that PTEN deficiency increases the resistance of cancer cells to TRAIL [266]. Therefore, these studies suggest that CP could restore TRAIL as a therapeutic option in these cancers, although this is yet to be discovered.

Whereas JNK and p38 MAPK promote p53 activity, the PI3K/Akt pathway in the cell inhibits p53 [227]. Consequently, aberrant Akt activity promotes cancer cell survival through inhibition of p53 [177,227]. Treating oral SCC with PAM has led to enriched p53 expression and, consequently, p21-mediated cell cycle arrest, inhibition of angiogenesis, DNA repair, and the mTOR (PI3k/Akt) pathways, and eventually caspase 8/9/3-driven apoptosis [209,267]. The ataxia telangiectasia mutated (ATM) pathway was also implicated in PAM- and direct CP-induced p53 activity [233,267], and although not cancer related, PAM selectively induced ATM expression in *Mycobacterium tuberculosis*-infected macrophages [268]. Additionally, the ATM pathway is associated with activity in a diverse array of other cell signalling pathways, including PI3K/Akt/mTOR [269]. Despite this, the ATM pathway response to CP and the relevance to anticancer effects of CP is still unclear. The multitude of consequences to deactivating PI3K/Akt signalling in cancer cells through cytotoxic CP exposure are summarised in Figure 6.

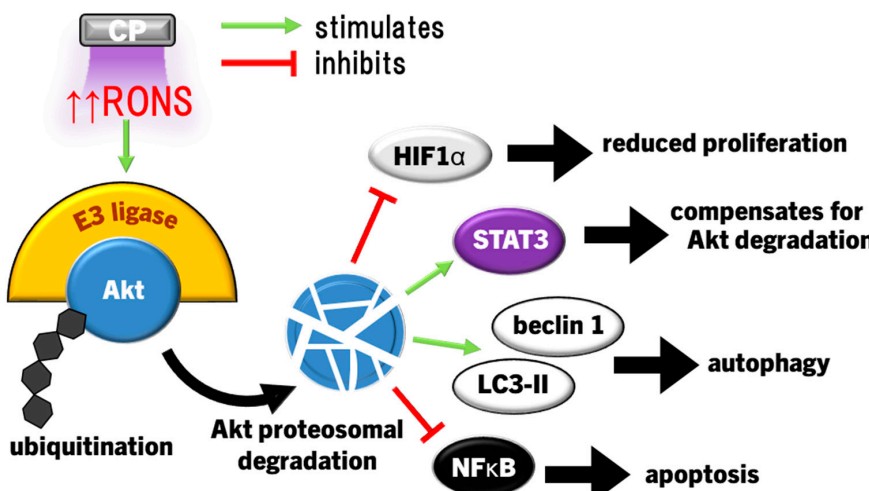

**Figure 6.** Complex physiological responses to deactivation of PI3K/Akt with high CP exposure. Excessive CP-derived RONS (indicated by "↑↑") leads to E3 ligase-dependent degradation of Akt, resulting in reduced hypoxia-inducible factor 1-alpha (HIF1α) signalling (antiproliferation), compensation by activation of signal transducer and activator of the transcription 3 (STAT3), particularly in PTEN-deficient cancer cells, activation of autophagic proteins beclin 1 and LC3-II, and inhibition of NFκB (apoptosis).

## 5. Discussion and Perspectives

As illustrated in this review, CP-derived RONS can be used to precondition normal cells against redox stress and promote pro-survival and wound healing by exploiting acute activation of Nrf2/ARE and UPR mechanisms [159,170–172,174,177]. Furthermore, "high" CP exposure can kill malignant cells through CHOP-dependent apoptosis as a result of prolonged UPR activity [151,211,212] and inhibition of Nrf2/ARE via Akt degradation and p53 [124,184]. Additionally, CP treatment could differentially modulate different MAPK pathways depending on the exposure time and whether the cells were malignant or non-malignant [39,123,147,164,217,233]. Similarly, CP has been shown to modulate PI3K/Akt in a bimodal fashion, with lower CP exposure times transiently activating Akt to promote survival, proliferation, and angiogenesis during tissue repair [143,253], while higher CP exposure times causing severe redox stress would lead to heavily suppressed or completely ablated Akt expression and activity, apoptosis, and autophagy [109,117,144,149,150,191]. The interactions between the intracellular signalling pathways and their effects on cell fate in response to CP exposure are summarised in Figure 7.

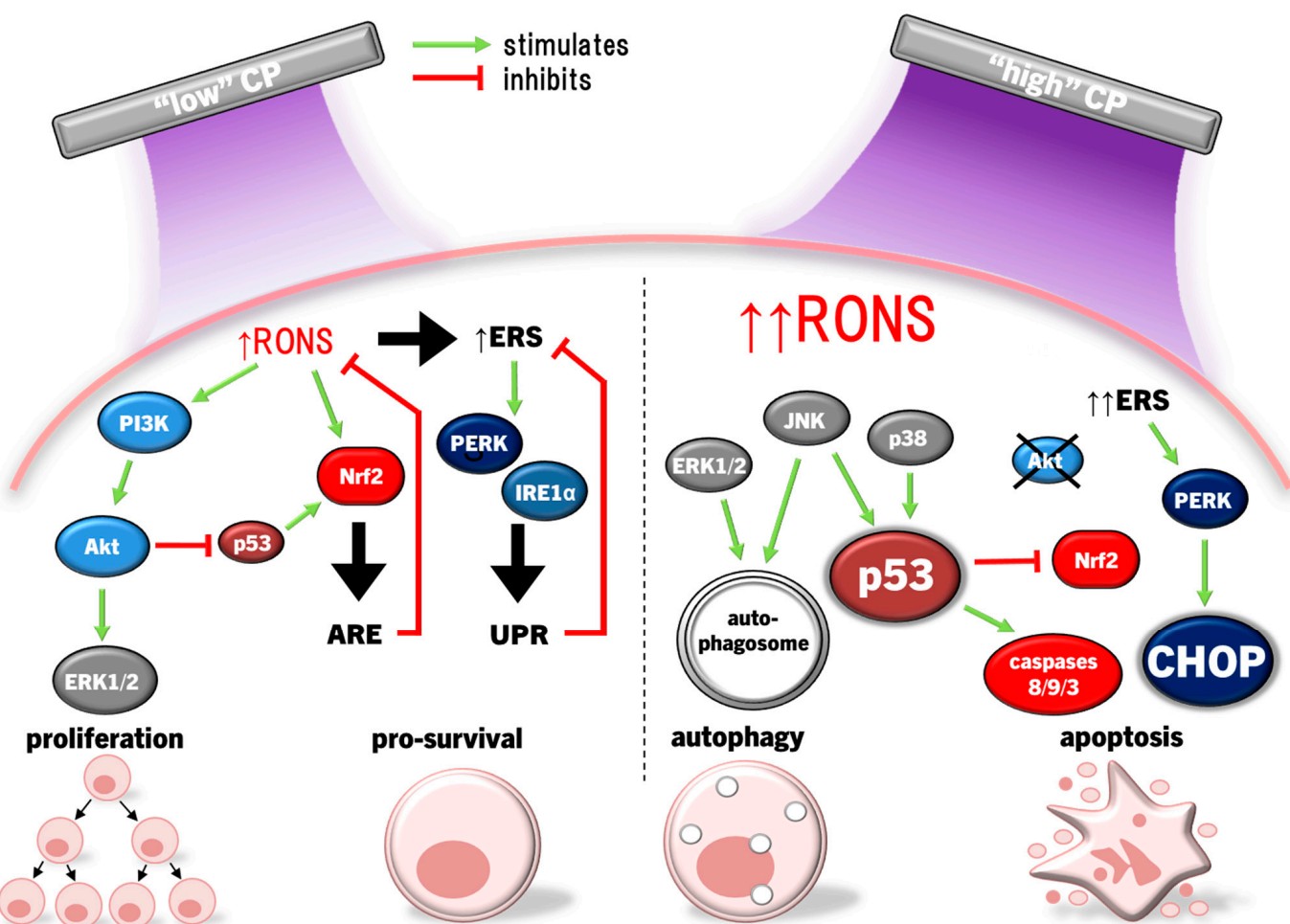

**Figure 7.** Overview of the effect of CP on cell signalling pathway activity, interactions between pathways, and cell fate decisions. The black cross indicates degradation of Akt. Black and red arrows indicate the direction of activation, being (↑) upregulated and (↑↑) prolonged or excessive upregulation.

### 5.1. Direct Comparisons between Malignant Cells and between Malignant and Non-Malignant Cells and Tissue

Current evidence shows that the most well-characterised responses to CP are related to anticancer treatment, bringing a wealth of knowledge for how CP causes antiproliferative, pro-apoptotic, and pro-autophagic responses in cancer cells. Several studies have directly

compared malignant and normal human cell responses to CP and PAL under constant or very similar cell culture conditions [84,115,116,140,144,164,226,229,232,261,270–274]. Studies have even compared the responses of multiple cell lines of the same cancer type or different cancers using different conditions for each cell line to investigate CP sensitivity [39,142,189,267]. Although useful insight may still be garnered, it is problematic to compare responses between cell types, particularly under different culture media conditions, and may lead to inaccurate conclusions. For instance, the selectivity of indirect CP treatment for inducing apoptosis in malignant cells and not healthy cells was only observed when cells were in their recommended media, but the selectivity was significantly diminished when cultured in the same media [271]. Similarly, comparing results from similar studies can also be problematic if the studies used different CP sources, making any comparisons mostly attributable to the difference in CP configuration. Accordingly, selectivity of CP and PAL can only be achieved after careful control of cell culture conditions between malignant and non-malignant cells. In saying this, only studies that compared malignant and healthy cell responses to CP under comparable cell culture and CP conditions are discussed here.

Skin keratinocytes, namely HaCaT, are the most utilised reference cell line for comparison with malignant cells. One recent study compared the cellular responses of HaCaT with SCC cells following indirect CP treatment under the same culture conditions and found drastically different responses. Compared to SCC cells, only HaCaT were shown to have upregulated expression of Nrf2-related proteins (NQO1, thioredoxin transmembrane protein 2, and GST M3) following indirect CP treatment [272]. Furthermore, HaCaT demonstrated increased cell motility and adhesion protein expression of angio-associated migratory cell protein (AAMP), Rho-associated coiled-coil containing protein kinase 2 (ROCK2), cortactin-binding protein 2 (CTTNBP2), cilia- and flagella-associated protein 20 (CFA20), and integrin-linked kinase-associated serine/threonine phosphatase 2C (ILKAP; which implicates β-catenin/Wnt signalling that affects cell migration [275]) following indirect CP treatment. In contrast, the same proteins are inversely regulated in SCC (HNO97) cells [272]. This was supported by other studies comparing HaCaT to melanoma cell lines and human brain cancer (U87) cells to normal astrocytes, showing that direct CP selectively killed cancerous, but not healthy cells by intracellular ROS inducing autophagy or p38- or JNK-dependent caspase 3-driven apoptosis [26,226,232,274]. PAM also upregulated hyaluronan synthase 3 (HAS3) expression that stimulated hyaluronan production in HaCaT cells, which promotes cell motility, adhesion, and growth [276], but not in A431 epidermoid squamous carcinoma cells [277]. Osteogenic responses to CP associated with p38 MAPK activation and GST antioxidant activity were also more pronounced in periodontal bone-derived stem cells than in bone-marrow-derived mesenchymal stem cells under the same conditions [278]. Evidently, there is strong in vitro support showing that CP can elicit healing activities in normal cells while cancer cells are either deactivated or killed. While these data are exciting, more mechanistic studies are required to confirm the suggested cancer selectivity of indirect or direct CP and PAL treatment on a cancer-specific basis.

Compared to normal prostate epithelial cells, prostate cancer cells had more stark JNK activation and additionally activated p38 to augment apoptosis [164]. Interestingly, quenching single RONS during CP exposure did not affect cancer cell cycle arrest, but quenching $H_2O_2$ or $O_3$ returned cell cycles to control levels in normal prostate epithelial cells [164]. Similarly, ERK1/2 autophagic responses (increased bax/bcl2 ratio and LC3 expression) to CP treatment in murine melanoma cells were not observed in normal cells [188]. Altogether, these results support the paradigm that cancer cells are more susceptible to redox stress than normal cells during CP treatment. SCC tissue and proximal healthy tissue also showed different responses to direct CP ex vivo, with higher levels of cytochrome c, higher IL10, TNFα, and IFNγ release, and lower IL22 released from tumour tissue [272] indicating significantly increased stress and apoptosis in malignant tissues. While these findings are promising, further studies are required to facilitate the translation of understanding from in vitro findings to animal and clinical testing.

Non-cancerous malignant and healthy cells have also been compared following CP treatment. For instance, there has been comparison to CP treatment between normal primary fibroblasts and fibroblasts from keloids; a benign, hyperproliferative, progressive fibrotic skin lesions characterised by overaccumulation of extracellular matrix proteins like collagen, with scarring that can be both physically and psychologically distressing for sufferers [279]. Interestingly, direct CP exposure did not kill either fibroblast type, but inhibited cell migration in keloid fibroblasts, associated with reduced collagen expression, suppressed ERK, Akt and STAT3 phosphorylation, while normal fibroblast migration and Akt phosphorylation was enhanced [280]. Even though this has currently been the only in vitro investigation into treating keloid with CP, an assessor-blinded, self-controlled trial of CP therapy for keloid was recently completed [281]. Eighteen participants had one randomised side of a keloid scar treated with a DBD CP device twice weekly for five weeks while the other side was untreated, and 30-day follow-up after the last treatment was conducted [281]. The colour, pigmentation, rubor, texture, and volume all steadily decreased after subsequent CP treatments, indicating keloid receding, with only mild scarring observed in one patient that quickly resolved itself [281]. Therefore, CP therapy could be a management strategy for abnormal tissue conditions. Unfortunately, as a keloid scar covering <5 cm$^2$ was treated for 5–15 min twice-weekly to marginally improve over several weeks [281], limitations with CP therapy will become apparent in cases when keloids (or other conditions) cover larger areas. CP has also shown efficacy against psoriasis, an inflammatory skin condition with abnormal keratinocyte activity that exhibit aberrant MAPK, PI3K/Akt, STAT3, and NFκB pathway [282]. Although the molecular pathway responses to CP in psoriasis are not well characterised, CP inhibits STAT3 activation and significantly remedy inflammatory and epidermal hypertrophic pathologies in a psoriasis mouse model [283,284].

*5.2. The Hormesis Principle to Plasma Medicine*

The mechanisms to utilise CP for therapeutic intervention can be divided into two categories based on the hormesis principle. The first is to induce oxidative eustress in tissue using "low" CP exposure to regulate local inflammatory signalling, bolster cellular redox signalling, and promote cell DNA/protein repair and antioxidant response mechanisms in healthy tissue. This promotes cell proliferation, leukocyte activation, angiogenesis, and tissue remodelling to accelerate healing. The second is applied to kill cancer cells using "high" CP exposure to overwhelm the antioxidant and DNA repair mechanisms mentioned above, inhibit proliferation, and promote apoptosis and autophagy. A hormesis framework for CP dosage in regard to wound healing, cancer therapy, and respective cell signalling modulation is represented in Figure 8.

What if cancer cells were instead stimulated and normal cells were lethally exposed to CP? Evidence to date would surmise that comparatively "low" doses of CP against cancer cells could be pro-malignant, while "high" doses of CP on wounds may slow down wound healing. The latter has been observed in full-thickness excisional wounds in rats, whereby 1 min CP exposure accelerated wound healing, while there was no difference between 3- or 5 min exposure to untreated wounds [241]. Correspondingly, lower CP exposure times stimulated tumour spheroid growth, until higher (≥120 s) exposure reduced spheroid growth again [191]. Additionally, osteosarcoma tumour growth was significantly elevated by PAL formed by 5 min CP activation compared to a slight decrease in tumour growth seen with PAL produced by 10 min CP exposure [191]. A significant external factor for the effectiveness of PAL was also the growth dimensions of cancer cells; 2D monolayers were significantly more susceptible to CP/PAL than 3D in vitro spheroids and tumours [191]. Therefore, there is a clear danger that must be recognised, namely, that CP exposures or PAL formulations that kill in vitro cultures of cancer cells may not be potent enough to impede tumour growth, or worse, may enhance tumourigenesis.

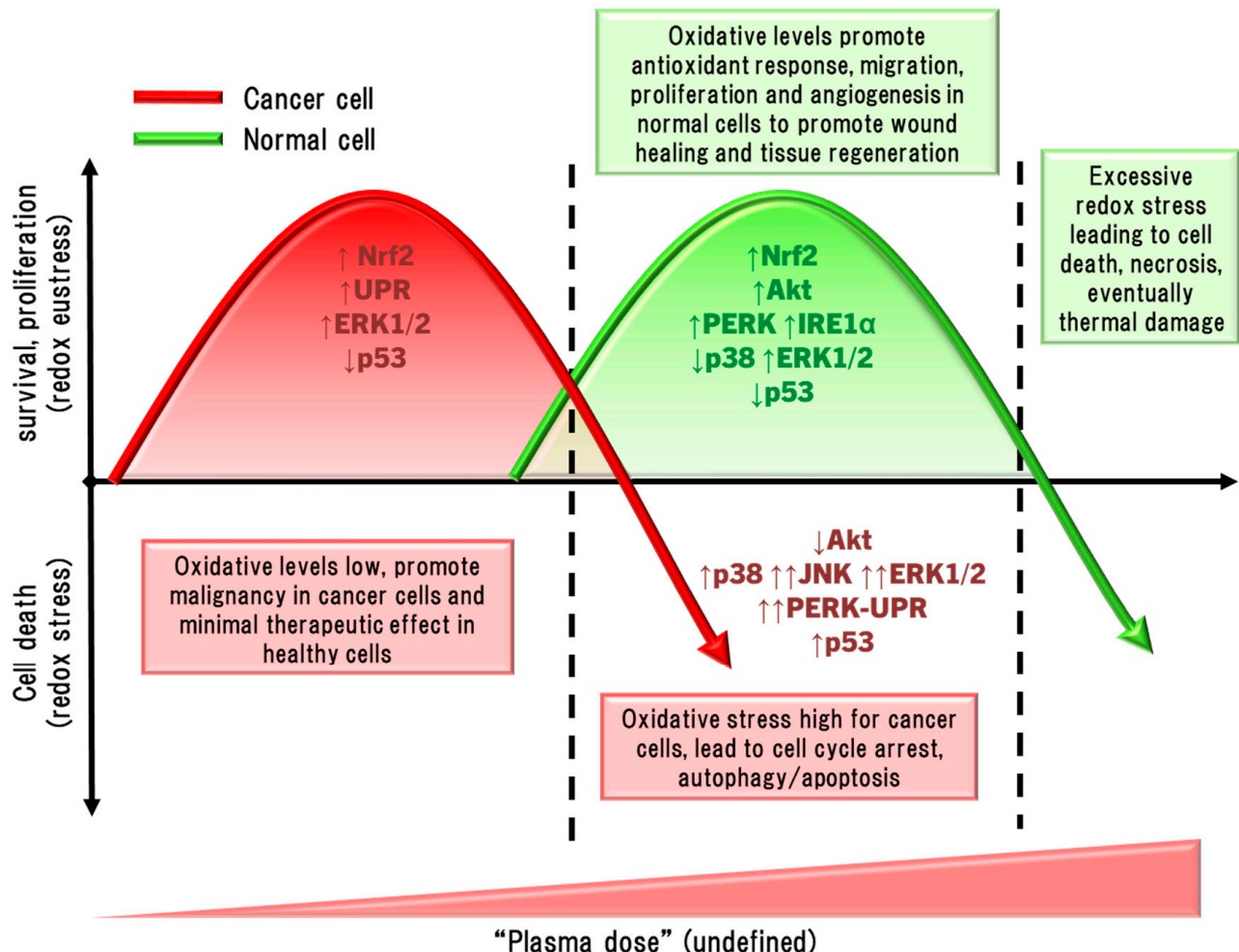

**Figure 8.** Conceptual hormesis response curves for cancer (red) and normal (green) cells, underpinned by the different responses in cell signalling pathway regulation between redox stress and eustress. The red and green text boxes represent cancer and normal cells, respectively. Arrows indicate the direction of activation, being (↑) upregulation, (↓) downregulation, and (↑↑) prolonged upregulation.

### 5.3. Limitations and Challenges in Plasma Medicine

There are several challenges that hinder understanding the biochemical and molecular pathways involved in the cellular response to CP. First, primary exposure to direct and indirect CP therapy usually lasts only seconds to several minutes, while downstream intracellular effects have only been investigated for hours to days after primary exposure ceased in most studies to date. Even so, what occurs in cells and tissues during active CP exposure is still largely unexplored. Difficulties in investigating biological responses in these short time scales have also been reported; an issue not unique to plasma medicine, but is highly relevant to the redox biology field. However, performing experiments using methods that will not interfere with RONS (or electric fields, radiation, and free electrons in regard to direct CP) is a significant technical challenge.

Secondly, research into cell signalling pathways has brought a wealth of knowledge on a variety of therapeutic targets of plasma medicine. However, a large majority of research is based on in vitro cell culture and comparing malignant to non-malignant cells. Some ex vivo experiments have shown limited direct clinical relevance to tissue-level response to CP therapy [132,174,177,272], but ex vivo experiments are often short in duration, are maintained in media that change its biological environment, and prohibit longer term experiments of CP therapy that are relevant to real application of CP on animals and

human subjects that could span weeks to months. Currently, in vivo investigations of the effects of CP on cell signalling pathways are well characterised in cancer models, but scarce in wound healing. Notably, only one animal model each has reported accelerated wound healing to ERK1/2 and Akt during CP treatment [241,254], with no other study to our knowledge supporting or contrasting this finding. No translation of the effect of CP inducing the UPR during early stages of wound healing has been performed either. Finally, there is a distinct lack of chronic wound models in animal research on CP therapy in wound healing that accurately replicate chronic wounds in humans [285].

Particularly problematic is trying to translate in vitro findings from PAL treatment, including PAM that surrogate for direct tissue exposure to CP. The effects of CP on media components are compounded by the different media compositions between studies that can drastically affect the RONS profile in the liquid [286] and the consequent downstream biological effects of CP in healthy and malignant cells [25,109,271]. Additionally, water itself is significantly cytotoxic in vitro owing to its hypotonicity, so PAW is usually diluted to half or quarter "concentration" in basal or growth media. This can be a major issue, as media can act as a buffer against pH, partially or significantly scavenge RONS of interest, and produce unknown or overlooked biologically active products, compromising the interpretation of the experimental results. For example, as stated earlier, the presence of pyruvate reduced SaOS-2 human osteosarcoma cell cytotoxicity to PAM by phosphorylation of ERK1/2, GSK3β, AMPK, and c-JUN and HSP60 dephosphorylation, which was primarily attributed to pyruvate scavenging $H_2O_2$ [109]. As pyruvate is a common media supplement as a carbon source for cell metabolism, especially in cancer malignancy [110], it is essential to consider its presence in the context of the plasma medicine experiments. This concept should be applied to other substrates/metabolites when possible. Therefore, experimental designs generally require additional control of variables like considering supplementation of preferably serum-free medium, excluding redox reactive supplements, and including exogenous RONS scavengers as additional treatment groups for validating presence of particular RONS. As such, interpretations for many of these experimental designs are limited to reasoning along the lines that PAM or the presence of PAW "contains RONS that may react directly with cells or respond to biologically active secondary redox products to induce/promote/suppress/ablate a cell response".

## 6. Concluding Remarks

At the cellular level, the effects of CP on Nrf2, UPR, PI3K/Akt, and MAPK signalling, their interplay, and downstream relationship with other tumour suppressor molecules like p53 and NFκB is well characterised. Translation to in vivo relevance is also emerging. Regardless, more research is needed into the role of Keap1 in response to CP beyond canonical Nrf2 inhibition.

As exemplified by the treatment of keloids and psoriasis, the initial phenotype of the cells, beyond whether the cell is normal or cancerous, may be a significant factor that influences the response to CP exposure. Therefore, other non-cancerous lesions and conditions that are caused by an adverse cell phenotype could be a viable target for CP therapy.

CP has already shown lack of genotoxicity in the short-term when applied to wound healing while still being potently antimicrobial to prevent and treat wound infections. Additionally, CP has potential as an adjunct therapy to synergise with chemo/radiotherapy to treat cancer. Furthermore, CP adjunct therapy may provide opportunities for cancer patients to receive lower chemotherapy doses to better control side-effects and thus improve quality of life in sufferers. Ideally, CP therapy may be applied to kill tumour cells at the margins of tumour resections and also stimulate healing of proximal healthy tissue to close wounds by the principles of hormesis and bystander effect. However, this dual effect, if possible, faces tremendous challenges. In particular, quantifying plasma "dose" will become imminently important, as clinical trials in plasma medicine mature into routine clinical use and more research is needed to best standardise reporting and clinical guidelines.

Further RCTs will help unravel the mechanism underpinning CP effects on wound healing and cancer, with hope of developing more personalised CP treatment with high efficacy and minimal side effects. Additionally, continuously monitoring the long-term prospective genotoxic potential of CP technology is required to be in step with progress in the field.

**Author Contributions:** Conceptualization, writing—original draft preparation, writing—review and editing, visualization, project administration, funding acquisition, A.I.A. writing—original draft preparation, writing—review and editing, funding acquisition, supervision Z.K. All authors have read and agreed to the published version of the manuscript.

**Funding:** Our research has been supported by the Australian Medical Research Future Fund (MRF2023153) and The Hospital Research Foundation. Z.K. is supported by the Channel 7 Children's Research Foundation Fellowship.

**Acknowledgments:** We would like to thank the multitude of plasma physicists, engineers, clinicians, health scientists, biochemists, molecular biologists, and chemists whose research has dramatically accelerated the field of molecular biology in plasma medicine. We also apologise to the authors whose important work could not be included in this review. Thank you to Laurine Kaul for proofreading and providing useful feedback and discussion.

**Conflicts of Interest:** The authors declare no conflicts of interest.

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
