# Peer review of "Comparing Redox and Intracellular Signalling Responses to Cold Plasma in Wound Healing and Cancer"

_cimb, doi:10.3390/cimb46050294_

Round 1

Reviewer 1 Report

Comments and Suggestions for Authors

The plasma treatment applications, in medicine presented in this work, is a distinctly up-to-date topic and a good approach for developing new therapeutic methods for wound healing and cancer treatment. The manuscript Comparing Redox and Intracellular Signalling Responses to Cold Plasma in Wound Healing and Cancer is an extensive review aiming to systematize studies in the field of plasma medicine applied in the treatment of wounds and cancer and to present the researched mechanisms for the response of biological objects to plasma treatment. The work done is consistent and the manuscript is well structured. This work will be highly appreciated in the circles of both plasma physicists and medical professionals.

I would like to advise the authors to rewrite the definition of plasma in the introduction. At the beginning of the text to omit the definition “Plasma is the fourth state of matter”, as it is controversial and a subject of disagreement among plasma physicists. Due to the lack of a phase transition between the gas state and plasma, the more generally accepted definition is “Plasma is an ionised gas consisting of ions, excited partials, free radicals and electrons that emits ultraviolet light”.

In the classification of plasma sources the authors describe two groups of plasma – Thermal plasma (as in the stars) and cold non-equilibrium plasma. The more precise classification is to divide plasma to Thermal (high temperature) and low temperature plasma. In turn, the low-temperature plasma is divided into non-equilibrium and equilibrium. The term “cold plasma” usually refers to a subtype of low-temperature plasma (non-equilibrium) with a particularly low temperature below 40°C that can be used for biomedical applications.

On line 41 “The large cool gas particles” are usually referred to as “heavy gas particles”.

On line 61 the gas flow for sustaining the plasma is usually referred to as “working gas” or “discharge gas”.

Of exceptional importance for the understanding of plasma medicine is the author’s statement that the findings discussed in this review should not be generalized to all CP sources designs. Taking into account that the consideration of different plasma sources is not the subject of this work, it remains unclear for the reader unfamiliar with plasma physics hence the significant difference when studying the effect of plasma treatment with different sources. The work would be greatly enriched by shortly stressing out that various plasma sources produce plasma with significantly different parameters such as active particles concentrations, RONS, temperature, electron densities and UV radiation.

At the clasification of the short-lived and long-lived RONS I would offer the authors two refferences that also clarify the importance of the treatment time duration:

Machala Z. et al., J. Phys. D: Appl. Phys., 52, 034002, 2019

P Lukes et al 2014 Plasma Sources Sci. Technol. 23 015019

On line 83 no definition and example of plasma-activated medium (PAM) has been given as well as plasma activated water (PAW).

In this part of the text also the easily applicability and storage time of PAW has been pointed as an advantage of its usage. It is important to mention that PAW storage time is questionable since the preservation of the properties and composition of the PAW is uncertain. Appropriate reference confirming the possibility of storage the PAL is needed.

On line 978 PAL obtained after different plasma treatment time is referred to as “weaker PAW” and “stronger PAW” when it will be more precisely to denote as a dependence of the treatment time or the concentration of active particles.

In part 5.1. a comment was made about the impossibility of comparing different studies due to non-uniformity of treated subjects. It would be good to include here that it is not appropriate to compare results obtained with different plasma sources, unless the goal is to find the advantages of one of them.

At the end of the work, the authors correctly discuss the impossibility of defining a plasma dose. This important comment should be made earlier in the text. Unfortunately, low and high exposure is used in the text referring for treatment time duration (line 708, 858, 74) and also as undefined “dose” (line 545, 567, 709, 860 and caption of fig. 3). On the other side “low” CP exposure and “height” CP exposure are used with the inaccuracy agreement in quotation marks (line 956, 545 and fig. 7). I would like to advise the authors to use “short and long plasma treatment time” wherever it is applicable in order to make it more understandable, and “low” CP exposure and “height” CP exposure when it is not.  

Reviewer 2 Report

Comments and Suggestions for Authors

Authors Abdo and Kopecki provide a comprehensive review manuscript about the application of Cold Atmospheric Plasma (CP) in wound healing and cancer. The structure of the manuscript is well orgnized, and the content is well informative. Several pathways were highlighted, including Keap1/Nrf2, ERS and UPR, MAPK and autophagy,  β-catenin, and PI3K/Akt. 

The only comment that I have is the cancer part. I agree with that there are common patten in redox state in caners, however, the redox state and antioxidant pathways are largely diverged progression and type of cancer. It would be helpful if authors could offer their opinions on this.
